# Cameras as Relative Positional Encoding

**Ruilong Li**[1,2*]   **Brent Yi**[1*]   **Junchen Liu**[1*]   **Hang Gao**[1]   **Yi Ma**[1,3]   **Angjoo Kanazawa**[1]

[1]*UC Berkeley*      [2]*NVIDIA*      [3]*HKU*

## Abstract

Transformers are increasingly prevalent for multiview computer vision tasks, where geometric relationships between viewpoints are critical for 3D perception. To leverage these relationships, multiview transformers must use camera geometry to ground visual tokens in 3D space. In this work, we compare techniques for conditioning transformers on cameras: token-level raymap encodings, attention-level relative pose encodings, and a new relative encoding—Projective Positional Encoding (PRoPE)—that captures complete camera frustums, both intrinsics and extrinsics, as a relative positional encoding. Our experiments begin by showing how relative conditioning methods improve performance in feedforward novel view synthesis, with further gains from PRoPE. This holds across settings: scenes with both shared and varying intrinsics, when combining token- and attention-level conditioning, and for generalization to inputs with out-of-distribution sequence lengths and camera intrinsics. We then verify that these benefits persist for different tasks, stereo depth estimation and discriminative spatial cognition, as well as larger model sizes. Code is available on our project webpage[2].

## 1   Introduction

Images of our world exist in the context of the viewpoints they were captured from. The geometry of these viewpoints—intrinsic and extrinsic parameters that give pixel coordinates their physical meaning—ground visual observations in 3D space. This spatial grounding is increasingly important in deep learning, especially as advances in 3D vision and embodied intelligence make multiview tasks more ubiquitous.

To solve multiview tasks with transformers, models must bind viewpoint information to patch tokens from each input image. This binding requires special care: just as naive positional encoding techniques for 1D sequences hinder performance for learning in language models [1], naive encodings of camera geometry may also be suboptimal for multiview vision models [2–4]. Advances in both settings can be summarized as transitions from absolute [5] to relative [6] encodings.

In this work, we study the problem of conditioning vision transformers on the camera geometry of input images. We survey existing techniques for addressing this, which include (i) absolute encodings in the form of pixel-aligned, token-level raymaps—these are the most common in recent state-of-the-art models [7–10]—and (ii) attention-level relative encodings based on SE(3) pose relationships [3, 4]. We then present a new camera conditioning technique, Projective Positional Encoding (PRoPE), that is designed to capture the complete geometry of cameras as a relative positional encoding. PRoPE models viewing *frustum* relationships that describe both intrinsics and extrinsics, while remaining easy to incorporate with standard transformer architectures and fused attention kernels [11].

Our experiments are on three tasks, which span six datasets. We begin with a series of studies comparing camera conditioning techniques for feedforward novel view synthesis using RealEstate10K [12] and Objaverse [13]. Our results highlight the advantages of relative encodings—particularly PRoPE—compared to absolute ones. We then verify that these benefits extend to other settings: we demonstrate improvements when integrating PRoPE into UniMatch [14] for stereo depth estimation across three

---

*Equal contribution

[2]https://www.liruilong.cn/prope/

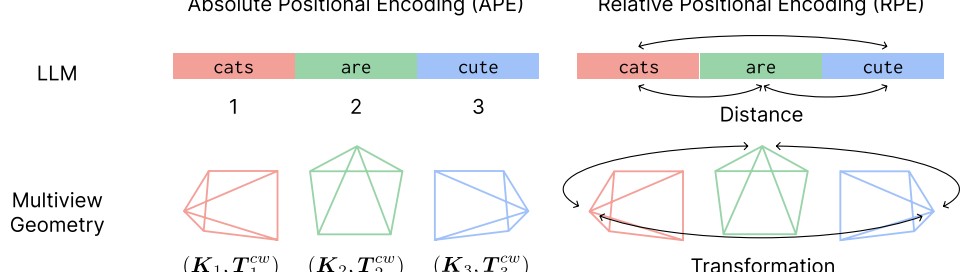

Figure 1: **Cameras as relative positional encoding.** Language models and multiview transformers must both bind "positional" information to input tokens, in terms of sequence position for language and camera parameters for multiview computer vision. We present a study on camera conditioning that includes absolute positional encodings (raymaps), relative pose encodings [2, 4], and a new method captures relative projective relationships between more complete camera *frustums*.

benchmarks, for a discriminative spatial cognition task using DL3DV [15], and when scaling to larger novel view synthesis models [7, 8].

The contributions of this paper are as follows:

**(1) Survey.** We survey both absolute raymap and relative SE(3) conditioning techniques for camera geometry in multiview transformers.

**(2) Method.** We propose PRoPE (Projective Positional Encoding), a new relative positional encoding technique that injects both camera intrinsics and extrinsics into a transformer's self-attention blocks.

**(3) Evaluation.** We present a series of novel view synthesis (NVS) experiments that compare camera conditioning techniques empirically. Our results highlight the advantages of relative pose encoding methods like CAPE [4] and GTA [3], while demonstrating further improvements from PRoPE across a range of settings: scenes with shared intrinsics, scenes with varying intrinsics, for hybrid conditioning that combines both token-level and attention-level representations, and in generalization for out-of-distribution test inputs.

**(4) Task generalization.** We show that the benefits of cameras as relative positional encoding generalize (i) to stereo depth estimation when integrated into UniMatch, (ii) to discriminative spatial cognition, and (iii) when scaling to larger model sizes.

## 2 Related Work

**Absolute and relative positional encodings.** Transformer architectures are permutation-invariant; they therefore require explicit position encoding to understand token order in sequential inputs [5]. Position encoding in sequence models has been an active area of research [1, 16–19]. While early works [20–27] focused on absolute positional encoding (APE), recent methods have increasingly adopted relative positional encoding (RPE), particularly RoPE [28], as a standard across domains, including natural language processing [29, 30, 1, 31] and computer vision [32–34, 28, 35]. Relative encodings aim to improve models by defining positions as relative offsets between token pairs. These offsets are injected into the pairwise interactions of standard dot product attention [5]:

$$\text{Attn}(Q, K, V) = \text{softmax}\left(\frac{QK^\top}{\sqrt{d}}\right)V, \tag{1}$$

where $Q, K, V \in \mathbb{R}^{T \times d}$. Position offsets can be injected using the pairwise nature of the $QK^\top \in \mathbb{R}^{T \times T}$ matrix, either via additive biases [6, 36, 37] or SO(2)-based rotation [1]. RPE offers important advantages over APE, including translation invariance, improved relationship modeling, and generalization to long sequences [38, 1, 39]. In this work, we study both absolute and relative encodings for conditioning transformers on camera geometry instead of 1D position.

**Multiview transformers.** Many computer vision tasks are multiview: they take multiple images and known camera geometry for each image as input. Examples exist in 3D reconstruction and view synthesis [40–44], pose estimation [45], depth prediction [14, 46, 47], 3D scene understanding [48, 49], robotics [50], and world models [51]. Many recent works leverage the improved scaling properties [22, 52, 53] of vision transformers for solving these tasks [8, 7, 9, 10, 54]. These models slice input images into patches, and use each patch as an independent visual token for the transformer. In this work, we use the model designs proposed by LVSM [8] and UniMatch [14] as a starting point for studying a critical design decision—how transformers are conditioned on camera geometry.

**Camera conditioning in transformers.** The dominant approach for conditioning multiview transformers on cameras is currently raymaps [45, 7, 8, 10]: per-pixel 6D embeddings that contain either ray origins and directions [44, 7] or Plücker coordinates [55, 45]. Concatenating these parameters to pixels enables conditioning on both camera intrinsics and extrinsics at the token level. Raymaps, however, require defining a frame of reference [56, 7, 57], which is problematic because the choice of world coordinate system is arbitrary and can hinder generalization. While this problem can be partially addressed by normalizing poses [58, 57, 8], prior works have shown that a more fundamental fix is possible through relative parameters [2–4, 59]. Notably, attention-level encodings that capture relative SE(3) poses don't require defining a consistent global frame, are compatible with fused attention kernels [11], and have been shown to improve novel view synthesis performance [3, 4]. In the following section, we survey both absolute raymap and relative SE(3) methods for encoding camera geometry for transformers. We then propose a new relative encoding method, PRoPE, that encodes relationships between more complete camera *frustums*.

## 3 Conditioning Transformers on Cameras

### 3.1 Preliminaries

We study transformers that take $N$ images from known cameras as input:

$$\{(\boldsymbol{I}_i, \boldsymbol{K}_i, \boldsymbol{T}_i^{cw})\}_{i=1}^N, \tag{2}$$

where each $\boldsymbol{I}_i \in \mathbb{R}^{H \times W \times 3}$ is an image, $\boldsymbol{K}_i \in \mathbb{R}^{3 \times 3}$ is the camera intrinsics, and $\boldsymbol{T}_i^{cw} = (\mathbf{R}_i^{cw}, \mathbf{t}_i^{cw}) \in \mathrm{SE}(3)$ is the transformation for computing camera coordinates from world coordinates. The latter two terms encode the viewing frustum that corresponds to each image: intrinsics capture the shape and field-of-view of the frustum, while extrinsics capture position and orientation. Both intrinsics and extrinsics are encapsulated in the "world-to-image" projection matrix $\boldsymbol{P}_i \in \mathbb{R}^{3 \times 4}$:

$$\boldsymbol{P}_i = \begin{bmatrix} \boldsymbol{K}_i & \mathbf{0}^{3 \times 1} \end{bmatrix} \boldsymbol{T}_i^{cw}. \tag{3}$$

For notational convenience, these $3 \times 4$ projection matrices can be made invertible by lifting to $4 \times 4$ with the standard basis vector $\mathbf{e}_4 = (0, 0, 0, 1)^\top$. This transformation maps 3D world coordinates to a projective image space defined by the frustum of camera $i$. It can be used to compute 2D image coordinates from world coordinates:

$$\tilde{\boldsymbol{P}}_i = \begin{bmatrix} \boldsymbol{P}_i \\ \mathbf{e}_4^\top \end{bmatrix}; \quad \begin{bmatrix} \tilde{\mathbf{x}}_i \\ 1 \end{bmatrix} \propto \tilde{\boldsymbol{P}}_i \tilde{\mathbf{X}}_{\mathrm{world}}, \tag{4}$$

where $\tilde{\mathbf{x}}_i \in \mathbb{R}^3$ and $\tilde{\mathbf{X}}_{\mathrm{world}} \in \mathbb{R}^4$ are homogeneous coordinates in the image and world respectively. The inverse relationship can be used to compute ray directions in 3D space from 2D image coordinates. For homogeneous image coordinate $\tilde{\mathbf{x}}_i^{u,v} = (u, v, 1)^\top$,

$$\begin{bmatrix} \alpha \mathbf{d}_i^{u,v} \\ 1 \end{bmatrix} \propto \tilde{\boldsymbol{P}}_i^{-1} \begin{bmatrix} \tilde{\mathbf{x}}_i^{u,v} \\ 1 \end{bmatrix}, \tag{5}$$

where $\alpha \in \mathbb{R}$ is a scalar magnitude and $\mathbf{d}_i^{\mathbf{u},\mathbf{v}} \in \mathbb{S}^2$ is a unit-norm ray direction.

### 3.2 Pixel-aligned Camera Encoding

Token-level, pixel-aligned raymaps are the dominant method for encoding geometry in multiview transformers [7, 42, 10, 9]. Networks that employ raymaps concatenate images $\boldsymbol{I}_i \in \mathbb{R}^{H \times W \times 3}$ with per-pixel raymaps $\mathbf{M}_i \in \mathbb{R}^{H \times W \times R}$ along the channel dimension, which expands inputs to $\mathbb{R}^{H \times W \times (3+R)}$. There are two main approaches for computing these raymaps, which we refer to as "naive" and Plücker.

**Naive raymaps.** Naive raymaps [7] are computed as per-pixel origin and direction vectors:

$$\mathbf{M}_{i,\mathtt{Naive}}^{u,v} = \begin{bmatrix} \mathbf{o}_i \\ \mathbf{d}_i^{u,v} \end{bmatrix} \in \mathbb{R}^6 \quad \text{for } (u, v) \in [1, W] \times [1, H] \tag{6}$$

$$\mathbf{o}_i = -(\mathbf{R}_i^{cw})^\top \mathbf{t}_i^{cw}, \tag{7}$$

where each ray direction $\mathbf{d}_i^{u,v}$ is computed using $\tilde{\boldsymbol{P}}_i$ by following Equation 5.

**Plücker raymaps.** Plücker raymaps [45, 8] can be implemented by replacing the origin term in naive raymaps with a moment term:

$$\mathbf{M}^{u,v}_{i,\texttt{Plücker}} = \begin{bmatrix} \boldsymbol{o}_i \times \boldsymbol{d}^{u,v}_i \\ \boldsymbol{d}^{u,v}_i \end{bmatrix} \in \mathbb{R}^6 \quad \text{for } (u,v) \in [1, W] \times [1, H]. \tag{8}$$

This moment term makes the ray representation invariant to the choice of origin along the ray.

**Properties.** Raymaps offer a simple approach for conditioning on both camera intrinsics and extrinsics. An important drawback, however, is that they are *absolute*: similar to early position encoding techniques for 1D sequences [5], raymaps are expressed in global terms. They are sensitive to the arbitrary choice of reference frame, which can hinder generalization.

## 3.3 Relative SE(3) Encoding

To remove the need for a global reference frame, recent works have introduced *relative* encodings for SE(3) camera poses. Two existing approaches fall under this category: CaPE [4] and GTA [3]. Given images $i_1$ and $i_2$, both aim to condition networks on $\boldsymbol{T}^{cw}_{i_1}(\boldsymbol{T}^{cw}_{i_2})^{-1}$ using modified self-attention blocks. This captures dense relationships—how each camera pose is situated relative to every other camera pose—and makes networks invariant to how the world frame $w$ is defined.

**Notation.** To formalize the operations required for attention-level geometry encoding, we denote the batched matrix-vector product $\odot$, Kronecker product $\otimes$, and identity matrices $\mathbf{I}_k \in \mathbb{R}^{k \times k}$. We use $i$ for image/camera indices and $t$ for patch/token indices. $i(t)$ is the index of the image that patch $t$ belongs to. Rows of the $Q, K, V \in \mathbb{R}^{T \times d}$ matrices are subscripted $Q_t, K_t, V_t \in \mathbb{R}^d$. Batched matrix-vector products are defined as:

$$\mathbf{A} \in \mathbb{R}^{N \times d_1 \times d_2}, B \in \mathbb{R}^{N \times d_2} \implies (\mathbf{A} \odot B) \in \mathbb{R}^{N \times d_1} \tag{9}$$

$$(\mathbf{A} \odot B)_{nj} = \sum_k \mathbf{A}_{njk} B_{nk}. \tag{10}$$

We use $\mathbf{D} \in \mathbb{R}^{T \times d \times d}$ to denote batches of block-diagonal matrices, where individual matrices in $\mathbf{D} = [\mathbf{D}_1, \ldots, \mathbf{D}_T]$ are subscripted $\mathbf{D}_t \in \mathbb{R}^{d \times d}$.

**CaPE [4].** CaPE injects relative SE(3) pose by transforming the $Q$ and $K$ matrices before they are passed to self-attention. CaPE can be formalized using per-token block-diagonal matrices $\mathbf{D}^{\texttt{CaPE}}_t$, which is computed by diagonally repeating the camera extrinsics:

$$\mathbf{D}^{\texttt{CaPE}}_t = \mathbf{I}_{d/4} \otimes \boldsymbol{T}^{cw}_{i(t)} \tag{11}$$

Like RoPE [1], transformations are then applied to the $Q$ and $K$ matrices before self-attention. This can be encapsulated into an augmented self-attention block:

$$\text{Attn}^{\texttt{CaPE}}(Q, K, V) = \text{Attn}((\mathbf{D}^{\texttt{CaPE}})^\top \odot Q, (\mathbf{D}^{\texttt{CaPE}})^{-1} \odot K, V). \tag{12}$$

The effect of this is that each $Q^\top_{t_1} K_{t_2} \in \mathbb{R}$ dot product in $QK^\top \in \mathbb{R}^{T \times T}$ is replaced with:

$$Q^\top_{t_1} \mathbf{D}^{\texttt{CaPE}}_{t_1} (\mathbf{D}^{\texttt{CaPE}}_{t_2})^{-1} K_{t_2} \in \mathbb{R}, \tag{13}$$

where $\mathbf{D}^{\texttt{CaPE}}_{t_1}(\mathbf{D}^{\texttt{CaPE}}_{t_2})^{-1} = \mathbf{I}_{d/4} \otimes \left[ \boldsymbol{T}^{cw}_{i(t_1)}(\boldsymbol{T}^{cw}_{i(t_2)})^{-1} \right]$, thus conditioning outputs on relative pose.

**GTA [3].** GTA proposes a formulation for per-token transformations with a similar high-level goal as CaPE. GTA's attention variant transforms the $Q$ and $K$ matrices the same way, while proposing to also transform the $V$ matrix:

$$\text{Attn}^{\texttt{GTA}}(Q, K, V) = \mathbf{D}^{\texttt{GTA}} \odot \text{Attn}((\mathbf{D}^{\texttt{GTA}})^\top \odot Q, (\mathbf{D}^{\texttt{GTA}})^{-1} \odot K, (\mathbf{D}^{\texttt{GTA}})^{-1} \odot V). \tag{14}$$

This has the added benefit of injecting relative transformations into the attention operator's value aggregation. The attention layer output for each token $t_1$ becomes

$$\left[ \text{Attn}^{\texttt{GTA}}(Q, K, V) \right]_{t_1} = \sum_{t_2} \alpha_{t_1, t_2} \mathbf{D}^{\texttt{GTA}}_{t_1} (\mathbf{D}^{\texttt{GTA}}_{t_2})^{-1} V_{t_2}, \tag{15}$$

where $\alpha_{t_1, t_2}$ is a softmax score computed from the transformed dot product (Equation 13). GTA's experiments [3] compare several SE(3)-based formulations for $\mathbf{D}^{\texttt{GTA}}$, with and without the value matrix transformation. The best-performing methods include the value transform, and both SE(3) for camera pose and RoPE [1] for 2D patch position. Our experiments include GTA using these terms.

## 3.4 Projective Position Encoding (PRoPE)

We introduce a new relative positional encoding method in our study, which we call PRoPE. The core observation of PRoPE is that the SE(3) poses considered by existing relative encoding techniques are only a partial representation of camera geometry. Instead of relating each camera $i_1$ and camera $i_2$ with only their poses $\boldsymbol{T}_{i_1}^{cw}$ and $\boldsymbol{T}_{i_2}^{cw}$, PRoPE uses the projective relationship between full *frustums*:

$$\tilde{\boldsymbol{P}}_{i_1}\tilde{\boldsymbol{P}}_{i_2}^{-1}. \tag{16}$$

This $4 \times 4$ matrix can be interpreted as a transformation between the local projective spaces defined by each camera; it encodes the complete geometric relationship between camera views. As we will see in Equation 20, it also retains the key global invariance property of SE(3)-based relative encodings.

To implement PRoPE, we define a new set of $\mathbf{D}_t^{\texttt{PRoPE}} \in \mathbb{R}^{d \times d}$ matrices and use GTA-style attention (Equation 14) to inject them into transformer blocks. We design these matrices to (1) encode frustum relationships *between* cameras—this uses the projective relationship in Equation 16—and (2) encode relative patch positions *within* cameras—this follows GTA [3] and uses RoPE terms. These goals are achieved with complementary submatrices, each with shape $\frac{d}{2} \times \frac{d}{2}$:

$$\mathbf{D}_t^{\texttt{PRoPE}} = \begin{bmatrix} \mathbf{D}_t^{\texttt{Proj}} & \mathbf{0} \\ \mathbf{0} & \mathbf{D}_t^{\texttt{RoPE}} \end{bmatrix} \tag{17}$$

$$\mathbf{D}_t^{\texttt{Proj}} = \mathbf{I}_{d/8} \otimes \tilde{\boldsymbol{P}}_{i(t)} \in \mathbb{R}^{\frac{d}{2} \times \frac{d}{2}} \tag{18}$$

$$\mathbf{D}_t^{\texttt{RoPE}} = \begin{bmatrix} \text{RoPE}_{d/4}(x_t) & \mathbf{0} \\ \mathbf{0} & \text{RoPE}_{d/4}(y_t) \end{bmatrix} \in \mathbb{R}^{\frac{d}{2} \times \frac{d}{2}}. \tag{19}$$

In these definitions, $\text{RoPE}_{d/4}(\cdot)$ constructs $\frac{d}{4} \times \frac{d}{4}$ rotary embeddings [1] for $(x_t, y_t)$, which are the patch coordinates for token $t$.

## 3.5 Properties of PRoPE

PRoPE has several important properties, which become more evident when we expand the projective transformation:

$$\tilde{\boldsymbol{P}}_{i_1}\tilde{\boldsymbol{P}}_{i_2}^{-1} = \begin{bmatrix} \boldsymbol{K}_i & \mathbf{0} \\ \mathbf{0} & 1 \end{bmatrix} \boldsymbol{T}_i^{cw}(\boldsymbol{T}_j^{cw})^{-1} \begin{bmatrix} \boldsymbol{K}_j^{-1} & \mathbf{0} \\ \mathbf{0} & 1 \end{bmatrix}. \tag{20}$$

**Global frame invariance.** Redefining the world frame is equivalent to right-multiplying both $\boldsymbol{T}^{cw}$ $SE(3)$ terms, which is algebraically eliminated in Equation 20.

**Reduction to relative SE(3) attention.** For cameras with identity intrinsics, Equation 20 reduces to the relative $SE(3)$ transformations utilized in CAPE and GTA. These methods can be interpreted as a case of PRoPE where the intrinsic matrices are set to identity.

**Reduction to RoPE [28].** Equation 20 evaluates to identity for patches from the same image. For these token pairs, $\mathbf{D}_t^{\texttt{PRoPE}}$ contains only the RoPE terms used by single-image vision transformers.

# 4 Experiments

The goal of our experiments is to understand how camera conditioning techniques—including PRoPE—impact the performance of multiview transformers. To accomplish this, we present experiments comparing encoding strategies under several task and evaluation conditions.

## 4.1 Experiment Setup

We include metrics for several camera conditioning techniques—Naive and Plücker raymap encodings, CAPE [4], GTA [3], and PRoPE. In our experiments, GTA refers to the SE(3)+SO(2) variation studied by [3], where SO(2) refers to RoPE on patch positions. As discussed in Section 3.4, PRoPE adopts the self-attention mechanism and RoPE combination proposed by GTA. The primary difference between PRoPE and GTA is therefore the use of relative projective relationships instead of relative SE(3) relationships between cameras.

Our core experiments evaluate camera conditioning techniques using feedforward novel view synthesis (NVS). NVS is an ideal benchmarking task because it requires fine-grained geometric reasoning:

| Method | RealEstate10K [12] | | | Objaverse [13] | | |
|---|---|---|---|---|---|---|
| | PSNR↑ | LPIPS↓ | SSIM↑ | PSNR↑ | LPIPS↓ | SSIM↑ |
| Plücker Raymap | 20.48 | 0.209 | 0.622 | 21.44 | 0.159 | 0.851 |
| Naive Raymap | 20.54 | 0.210 | 0.623 | 21.59 | 0.153 | 0.856 |
| CAPE [4] | 21.11 | 0.234 | 0.656 | 19.68 | 0.220 | 0.827 |
| GTA [3] | 22.51 | 0.164 | 0.707 | **23.70** | **0.104** | **0.879** |
| PRoPE | **22.80** | **0.146** | **0.725** | **23.70** | **0.104** | **0.879** |

Table 1: **Novel view synthesis comparison, with constant intrinsics in each scene**. We compare different camera conditioning approaches applied to the LVSM [8] framework.

| Method | RealEstate10K [12] | | | Objaverse [13] | | |
|---|---|---|---|---|---|---|
| | PSNR↑ | LPIPS↓ | SSIM↑ | PSNR↑ | LPIPS↓ | SSIM↑ |
| Plücker Raymap | 19.89 | 0.327 | 0.608 | 21.43 | 0.177 | 0.852 |
| Naive Raymap | 20.56 | 0.301 | 0.629 | 21.00 | 0.191 | 0.845 |
| CAPE [4] | 15.94 | 0.497 | 0.699 | 16.78 | 0.322 | 0.760 |
| GTA [3] | 15.77 | 0.512 | 0.641 | 18.00 | 0.257 | 0.775 |
| PRoPE | **21.42** | **0.247** | **0.678** | **22.98** | **0.138** | **0.871** |

Table 2: **Novel view synthesis, with varying intrinsics in each scene**. We compare the LVSM [8] model trained with different camera conditioning strategies, on intrinsics-augmented dataset variants.

models are trained to render scenes from target viewpoints, given only calibrated reference images and target camera parameters. We do this by reimplementing and training variants of LVSM [8], a state-of-the-art novel view synthesis method that originally encoded camera geometry with Plücker raymaps. We train and evaluate separately on the RealEstate10K [12] and Objaverse [13] datasets. Scenes in RealEstate10K are captured with constant intrinsics, but cameras vary between scenes. Objaverse renders use the same camera intrinsics across the entire dataset.

We take several steps to ensure fairness in evaluations. Models are trained in the same codebase, with matched hyperparameters and training steps. All main experiments use identical input, output, and overall model sizes (∼25M parameters); we also validate larger models in Section 4.7. More details are provided in Appendix A.1.1.

## 4.2 Relative vs Absolute Positional Encodings

Novel view synthesis results are presented in Table 1 and discussed below.

**Relative encodings outperform absolute ones.** Consistent with prior work [4, 3], we observe that relative encodings for camera geometry consistently outperform absolute ones. CAPE, GTA, and PRoPE all yield improvements over widely used raymap encodings, with PRoPE (followed closely by GTA) producing the best results.

**Projective positional encoding improves view synthesis quality.** PRoPE consistently outperforms other encoding methods across metrics on RealEstate10K [12], despite the dataset's limited intrinsics variation. This confirms that capturing more complete camera information (both intrinsic and extrinsic) in our relative encoding is beneficial. We also find no loss of performance when the train and test images all have constant intrinsics: GTA [3] and PRoPE produce identical metrics for the Objaverse dataset, which verifies that PRoPE reduces to GTA when camera intrinsics are unimportant.

## 4.3 Attention-Level Intrinsics Conditioning

Real-world data often involves different cameras and focal lengths—consider multiview rigs on autonomous cars or zoom lenses on point-and-shoot cameras. To understand how effective PRoPE is at encoding intrinsics information, we evaluate each conditioning method on intrinsics-augmented versions of the RealEstate10K [12] and Objaverse [13] datasets. We augment RealEstate10K by applying a zoom factor sampled uniformly in $[1, 3]$ to each image. For Objaverse, we switch from constant field of views to uniformly sampled ones between 35 and 50 degrees. In contrast to Section 4.2, this means that cameras within scenes can vary in both extrinsics *and* intrinsics. We present quantitative results in Table 2 and qualitative results in Figure A.1.

**PRoPE enables intrinsic-aware multiview understanding.** We observe that PRoPE outperforms all alternative camera conditioning techniques for both the RealEstate10K and Objaverse datasets. Existing attention-based methods, which let networks condition on relative pose, perform poorly

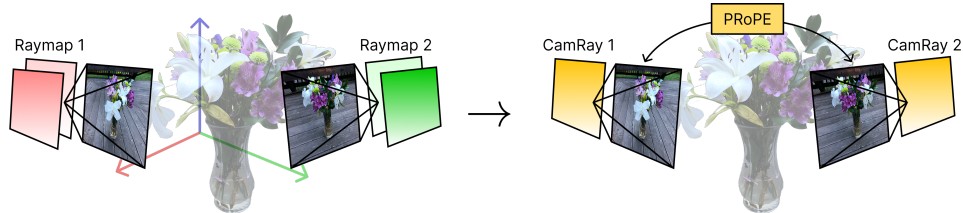

Figure 2: **Hybrid camera encoding.** *Left:* token-level conditioning using only raymaps, which capture both intrinsics and extrinsics. *Right:* hybrid encoding where attention-level PRoPE captures relationships between camera frustums, and token-level CamRay encodes local ray directions.

| Method | RealEstate10K [12] | | | Objaverse [13] | | |
|---|---|---|---|---|---|---|
| | PSNR↑ | LPIPS↓ | SSIM↑ | PSNR↑ | LPIPS↓ | SSIM↑ |
| GTA [3] | 15.77 | 0.512 | 0.641 | 16.98 | 0.261 | 0.762 |
| GTA+CamRay | 21.41 | 0.238 | 0.673 | 22.69 | 0.123 | 0.869 |
| PRoPE | 21.42 | 0.247 | 0.678 | 22.82 | 0.119 | 0.872 |
| PRoPE+CamRay | **21.78** | **0.211** | **0.692** | **22.98** | **0.114** | **0.874** |

Table 3: **Novel view synthesis with *hybrid* camera encodings, with varying intrinsics in each scene**. Intrinsics can be conditioned by concatenating local frame camera rays to the network input.

without knowledge of intrinsics. While token-level raymaps carry sufficient camera information, they perform worse overall than PRoPE's relative conditioning formulation.

## 4.4 Hybrid Encoding Strategies

Token- and attention-level camera encodings require modifications to different parts of the transformer architecture. They are therefore compatible with each other: both conditioning styles can be used simultaneously. To compare PRoPE against a stronger baseline while simultaneously exploring these "hybrid" conditioning strategies (Figure 2), we train LVSM variations that couple relative encodings with local, *camera-frame* raymaps:

$$\mathbf{M}_{i,\texttt{CamRay}}^{u,v} = \mathbf{R}_i^{\text{cw}} \mathbf{d}_i^{u,v} \propto \boldsymbol{K}_i^{-1} \begin{bmatrix} u & v & 1 \end{bmatrix}^\top \in \mathbb{R}^3 \tag{21}$$

We refer to this raymap as CamRay. CamRay shares many properties with existing raymaps (Section 3.2)—it encodes intrinsics, is pixel-aligned, and can be concatenated to input images—but it is not tied to an absolute coordinate system. It can therefore be used in conjunction with relative pose and camera encoding techniques without sacrificing global frame invariance. As we observe in Section 4.6, this provides empirical advantages over Plücker raymaps.

CamRay can be interpreted as a token-level encoding of camera intrinsics. We therefore evaluate it using the intrinsics-augmented NVS datasets described in Section 4.3. Results are reported in Table 3 and discussed below.

**PRoPE effectively encodes camera geometry.** On both RealEstate10K and Objaverse, we observe that PRoPE is comparable with or outperforms GTA+CamRay. This is true even though PRoPE is simpler: it is only applied attention-level, while GTA+CamRay includes both attention-level and token-level terms.

**Token-level and attention-level conditioning techniques are complementary.** GTA and PRoPE both benefit from the extra CamRay input. GTA benefits significantly more; this can be explained by the absence of intrinsics information in the standard SE(3)-based GTA formulation.

## 4.5 Out-of-distribution Robustness

One hypothesis for why relative camera encodings outperform absolute ones is improved generalization characteristics; this is similar to why RoPE [28] can improve performance for language modeling. To test this, we benchmark conditioning methods using test-time settings that introduce distribution shifts in sequence length and intrinsics (Figure 3).

*Setting 1: Longer Input Sequences at Test Time.* Inspired by test-time context length extrapolation [60, 61], our first setting deploys NVS models trained with a fixed number of input views (2 in our experiments) to significantly more views (up to 16) at test time. This is particularly important

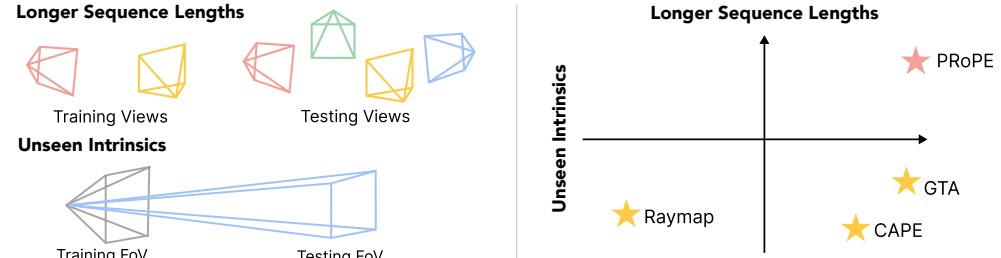

Figure 3: **Out-of-distribution tasks.** *Left:* We evaluate camera conditioning methods on both longer sequence lengths and unseen camera intrinsics. *Right:* PRoPE improves results for both unseen sequence lengths and unseen intrinsics.

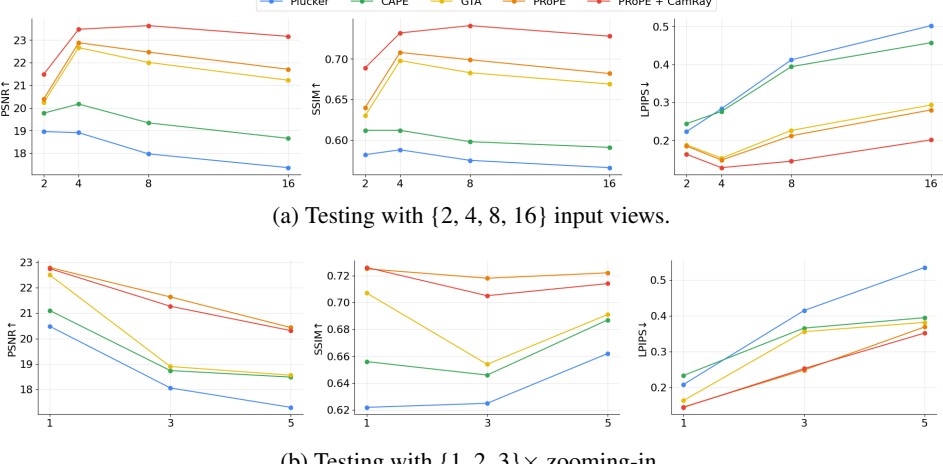

(a) Testing with {2, 4, 8, 16} input views.

(b) Testing with {1, 2, 3}× zooming-in.

Figure 4: **Evaluation on RealEstate10K** [12]. Relative encoding methods demonstrate superior robustness on handling (a) varying numbers of input views and (b) different focal lengths at test time.

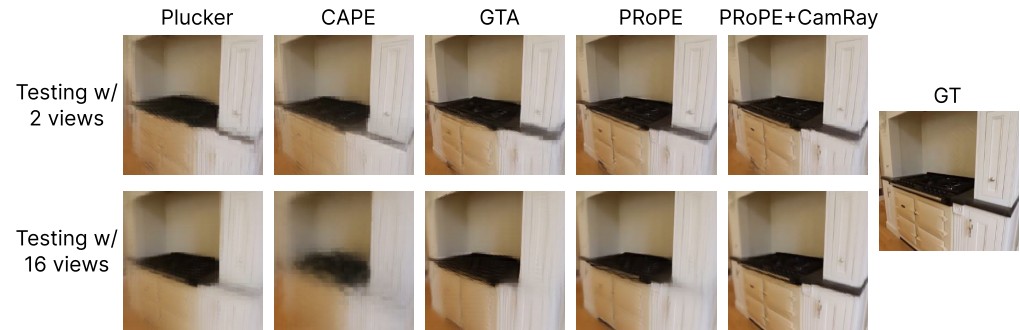

Figure 5: **Results of Longer Input Sequences at Test Time.** PRoPE demonstrates superior robustness when tested with varying numbers of input views, despite being trained with only 2 views.

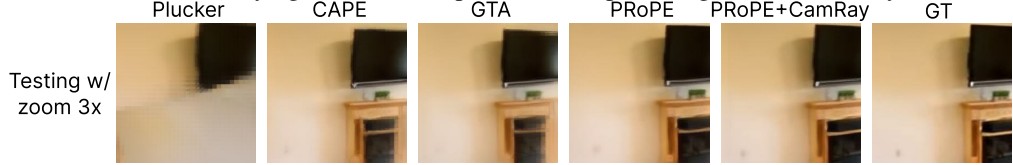

Figure 6: **Results of Varying Focal Length at Test Time.** PRoPE explicitly models relative intrinsics—we find this makes it more robust to zoomed-in test views.

in real-world scenarios where the number of observations can vary substantially across different applications, and sometimes dynamically increase.

*Setting 2: Out-of-distribution Intrinsics at Test Time.* Our second setting evaluates a model's ability to handle varying focal lengths at test time. This is crucial as focal length can vary significantly across

| Dataset | Model | Abs Rel | Sq Rel | RMSE | RMSE log |
|---|---|---|---|---|---|
| RGBD [62] | UniMatch | 0.123 | 0.175† | 0.678 | 0.203 |
| | UniMatch + PRoPE | **0.105** | 0.203† | **0.573** | **0.181** |
| SUN3D [63] | UniMatch | 0.131 | 0.098 | 0.397 | 0.169 |
| | UniMatch + PRoPE | **0.117** | **0.075** | **0.343** | **0.152** |
| Scenes11 [64] | UniMatch | 0.065 | 0.085 | 0.575 | 0.126 |
| | UniMatch + PRoPE | **0.049** | **0.063** | **0.474** | **0.104** |

Table 4: **Performance Improvement on Stereo Depth Estimation Task with UniMatch [14]**. †The "Sq Rel" metric is less reliable on the RGBD dataset due to the imperfect depth and camera pose [64].

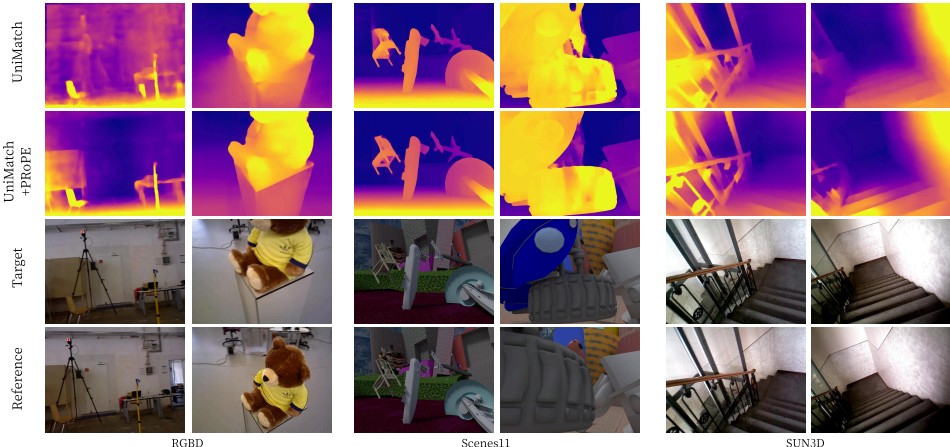

Figure 7: **Qualitative Results on Stereo Depth Estimation Task.** Attention-level camera conditioning in UniMatch [14] leads to significant estimation improvements.

| Method | 5 views | 9 views | 17 views |
|---|---|---|---|
| Plücker | 69.1% | 76.9% | 74.6% |
| PRoPE+Plücker | 81.1% | 90.5% | 91.8% |
| PRoPE+CamRay | **86.1%** | **93.0%** | **94.3%** |

Table 5: **Spatial cognition results**. We report the accuracy of detecting inconsistent image-camera pairs on the DL3DV [15] dataset under varying numbers of input views. Both CamRay and PRoPE significantly help with performance, without introducing additional model parameters. An illustration of this task can be found in Figure A.3.

different cameras and zoom levels, and it is impractical to train a separate model for every possible focal length. We test our models with focal lengths ranging from $1\times$ to $5\times$ the training focal length, simulating scenarios where more zoomed-in images are seen during deployment.

**Relative encodings improve generalization; PRoPE outperforms alternatives.** Evaluated results on the RealEstate10K [12] dataset are summarized in Figure 4, with visuals provided in Figure 5 and 6. We make three main observations. First, while Plücker Raymap encodes more complete camera information than CAPE and GTA, it consistently underperforms across all settings—even when intrinsics information is critical. Second, PRoPE improves performance and robustness in both out-of-distribution settings, particularly when handling out-of-distribution focal lengths. This indicates that explicitly modeling the relative projective relationship between cameras is more effective than modeling only the relative SE(3) relationship, as done in GTA and CAPE. Finally, we found adding CamRay to PRoPE actually *hurts* performance for intrinsics extrapolation; this suggests that PRoPE is uniquely useful for intrinsics generalization.

### 4.6 Task Generalization

Like our results so far, prior studies on relative pose encodings [3, 4] for multiview transformers have focused experiments on novel view synthesis. To better understand how these conclusions generalize, we evaluate PRoPE in two new tasks: stereo depth estimation using UniMatch [14] and a spatial cognition task designed around DL3DV [15].

*Stereo Depth Estimation.* We set up this task using UniMatch [14], a pretrained multiview transformer that has seen wide adoption in downstream applications [65–68]. UniMatch was originally trained on

| Method | 1× Compute | | | 100× Compute | | |
|---|---|---|---|---|---|---|
| | PSNR↑ | SSIM↑ | LPIPS↓ | PSNR↑ | SSIM↑ | LPIPS↓ |
| Plücker Raymap | 20.48 | 0.622 | 0.209 | 25.64 | 0.809 | 0.084 |
| PRoPE | **22.80** | **0.725** | **0.146** | **26.56** | **0.832** | **0.071** |

Table 6: **Scaling LVSM with increased compute.** We compare LVSM [8] models trained with Plücker raymaps versus PRoPE on RealEstate10K [12] at two compute scales.

three different tasks; we focus on the stereo depth estimation task, which assumes known relative camera poses between input views. We incorporate camera information into UniMatch's cross-view attention mechanism using PRoPE, modifying only ∼50 lines of the official code. All models follow the exact same training protocols as described in the original paper.

*Spatial Cognition.* Next, we design a spatial cognition task inspired by [69]. In this task, a network is given multiple images of the same scene, each paired with camera information. The problem is designed so that it cannot be solved by analyzing the camera information alone, the images alone, or without reasoning about the multiview relationships among all inputs. One of the image-camera pairs is intentionally corrupted by assigning it an incorrect camera pose sampled from other frames. The network is then required to identify the incorrect image-camera pair based on geometric consistency. See Appendix A.1.3 for implementation details, and Figure A.3 for input-output examples.

**PRoPE's benefits generalize across tasks.** For depth estimation, we provide quantitative results in Table 4 and qualitative results in Figure 7. For spatial cognition, accuracy metrics are provided in Table 5. We find that PRoPE significantly improves multiview understanding across both tasks. In our spatial cognition task, we observe that performance with PRoPE continues to improve as the number of views increase during testing, whereas the Plücker raymaps do not exhibit the same trend. We also observe improvements when replacing Plücker with CamRay, which indicates that the absolute extrinsics information hinders the model's ability to generalize.

### 4.7 Scaling PRoPE

In our final set of experiments, we evaluate how the advantages of relative camera encoding extend to larger models with more computational resources. We conducted two experiments for this, which are discussed below.

**PRoPE's benefits persist when scaling LVSM.** We scale our LVSM training pipeline with approximately 100× more computational resources (details in Appendix A.1.1). We train two variants of this larger LVSM model: one that follows the original LVSM paper [8] and uses Plücker rays and one that incorporates PRoPE. Results are reported in Table 6, where we observe that the relative PRoPE encoding continues to improve model quality—by a smaller but still significant margin—on larger model variants trained with more resources.

**PRoPE improves CAT3D results.** With assistance from the original authors, we add PRoPE to and retrain CAT3D [7], a large multiview diffusion model conditioned on naive raymaps. We report metrics from this model in Table A.2. PRoPE produces consistent improvements across metrics, while introducing zero additional model parameters and negligible computational overhead.

## 5 Conclusion and Future Work

In this work, we highlight how representing cameras as relative positional encodings—particularly in a way that captures both camera intrinsics and extrinsics—improves multiview transformers across settings and tasks. Possibilities for future work include improving numerical stability when directly multiplying projective matrices with Q/K/V vectors; it may be possible, for example, for ill-conditioned matrices to emerge from telephoto focal lengths. It would also be interesting to extend PRoPE to distorted camera models, for example by applying PRoPE to per-patch projective approximations. Furthermore, incorporating "multi-frequency" [19, 1] encoding for camera parameters remains nontrivial due to the non-commutativity of projective transforms, a challenge shared by relative SE(3) attention methods [3, 4].

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

## A.1 Experiment details

### A.1.1 LVSM-based Model Details

We adhere to the original LVSM [8] implementation specifications, maintaining consistency across most settings. For complete details regarding these configurations, we direct readers to the original LVSM paper [8]. The modifications we made include:

- We trained exclusively at 256×256 resolution and did not perform the additional fine-tuning at higher resolutions.

- Limited by academic-level resources, we use a smaller version of the LVSM model with 6 transformer blocks, and reduce the MLP channel dimension from 3072 to 1024. Our models are trained on 2x GPUs with a total batch size of 4, as opposed to 512 in the original paper. This applies to all experiments expect for the ones in Section 4.7.

- For the scaling-up experiments in Section 4.7, we use a LVSM model with 12 transformer blocks and keep the all other configurations including the MLP channel dimension (3072). These models are trained on 8x GPUs with a total batch size of 64.

### A.1.2 UniMatch Modification Details

UniMatch [14]'s model consists of a cross-attention transformer to capture multi-view relationship, as well as a self-attention transformer that serves as a single-view image encoder/decoder. We inject the camera information into both transformers, on the Q/K/V/O vectors, using our formulation. Note that our formula exactly falls back to RoPE [28] in the single-view scenario. It therefore naturally works with both transformer networks in UniMatch.

### A.1.3 Spatial Cognition Model Details

We formulate this task as a classification problem, where the number of classes corresponds to the number of input image-camera pairs, and the "ground-truth" class is the ID of the inconsistent image-camera pair. The training objective is to identify the inconsistent pair, which we optimize using the cross-entropy loss. The architecture of the model is largely similar to LVSM [8], where the only change is that the last linear layer is modified to output a single scalar for each token. Outputs are then averaged over all tokens for each input pair, resulting in a vector that represents the probability that each input pair is the "bad" one (over softmax). We provide more data exemplars we used for training and testing in Figure A.3.

## A.2 Additional Results

### A.2.1 Ablating PRoPE

As our proposed PRoPE includes two terms: the projective relationship between cameras ($\mathbf{D}_t^{\text{Proj}}$) and the patch coordinate relationship ($\mathbf{D}_t^{\text{RoPE}}$). We here ablate each of them to study their contribution. As shown in Table A.1, as a crucial component to the system, modeling the projective relationship between cameras alone already yields strong performance. Introducing RoPE [28] further enhances model's ability on understanding patch relationship, which is a minimal unit in vision transformer architecture. Notably, in PRoPE, we allocate half of the feature channels to encode each term. Ablating one term therefore means using all feature channels to encode the remaining one.

### A.2.2 PRoPE v.s. GTA: More Analysis on the Affect of Intrinsic Modeling

The main difference between our proposed PRoPE and prior work GTA [3] is the introduce of camera intrinsic into the formulation. While GTA itself does not take into account camera intrinsics, as we have pointed out in Section 4.4, it is compatible with our proposed CamRay – which encodes the intrinsic at token level – to form a complete representation for camera conditioning. In this section, we run more thorough experiments focusing comparing PRoPE and GTA in differnet settings (with/without CamRay) as the camera conditioning techniques on both UniMatch for stereo depth

| Encoding Method | PSNR↑ | LPIPS↓ | SSIM↑ |
|---|---|---|---|
| w/o $\mathbf{D}_t^{\text{Proj}}$ | 16.04 | 0.505 | 0.509 |
| w/o $\mathbf{D}_t^{\text{RoPE}}$ | 21.39 | 0.238 | 0.673 |
| PRoPE | **21.78** | **0.211** | **0.692** |

Table A.1: **Ablation Study on PRoPE.** $\mathbf{D}_t^{\text{Proj}}$ is crucial for encoding the relative camera information, and $\mathbf{D}_t^{\text{RoPE}}$ is also helpful to capture the relative patch coordinate. Experiments are conducted on RealEstate10K [12] with CamRay as input.

| Method | 3-view | | | 6-view | | | 9-view | | |
|---|---|---|---|---|---|---|---|---|---|
| | PSNR ↑ | SSIM ↑ | LPIPS ↓ | PSNR ↑ | SSIM ↑ | LPIPS ↓ | PSNR ↑ | SSIM ↑ | LPIPS ↓ |
| Zip-NeRF [70] | 12.77 | 0.271 | 0.705 | 13.61 | 0.284 | 0.663 | 14.30 | 0.312 | 0.633 |
| ZeroNeRF [71] | 14.44 | 0.316 | 0.680 | 15.51 | 0.337 | 0.663 | 15.99 | 0.350 | 0.655 |
| ReconFusion [72] | 15.50 | 0.358 | 0.585 | 16.93 | 0.401 | 0.544 | 18.19 | 0.432 | 0.511 |
| CAT3D [7] | 16.62 | 0.377 | 0.515 | 17.72 | 0.425 | 0.482 | 18.67 | 0.460 | **0.460** |
| CAT3D [7] + PRoPE | **16.93** | **0.382** | **0.505** | **18.01** | **0.443** | **0.479** | **18.98** | **0.474** | 0.461 |

Table A.2: **Incorporating PRoPE into CAT3D.** Adding PRoPE to the CAT3D [7] multiview diffusion model yields improvements over the original model, with zero additional model parameters and negligible computational overhead.

| Method | 1× | 3× | 5× | 7× |
|---|---|---|---|---|
| Plücker | 19.89 | 21.03 | 20.75 | 20.32 |
| GTA | 15.77 | 18.14 | 18.12 | 18.29 |
| GTA+CamRay | 21.41 | 22.70 | 22.35 | 21.77 |
| PRoPE | 21.42 | 22.95 | 22.75 | **22.40** |
| PRoPE+CamRay | **21.86** | **23.06** | **22.83** | **22.40** |

PSNR (↑) across zoom-in levels.

| Method | 1× | 3× | 5× | 7× |
|---|---|---|---|---|
| Plücker | 0.608 | 0.694 | 0.732 | 0.755 |
| GTA | 0.512 | 0.626 | 0.683 | 0.721 |
| GTA+CamRay | 0.673 | 0.741 | 0.764 | 0.780 |
| PRoPE | 0.678 | 0.748 | 0.775 | **0.794** |
| PRoPE+CamRay | **0.693** | **0.751** | **0.776** | **0.794** |

SSIM (↑) across zoom-in levels.

| Method | 1× | 3× | 5× | 7× |
|---|---|---|---|---|
| Plücker | 0.327 | 0.272 | 0.308 | 0.345 |
| GTA | 0.641 | 0.488 | 0.439 | 0.403 |
| GTA+CamRay | 0.238 | 0.191 | 0.242 | 0.288 |
| PRoPE | 0.247 | 0.193 | 0.227 | 0.244 |
| PRoPE+CamRay | **0.218** | **0.183** | **0.220** | **0.242** |

LPIPS (↓) across zoom-in levels.

Table A.3: **Additional Novel View Synthesis Results on Out-of-distribution Intrinsics at Test Time.** Experiments are conducted with LVSM [8] on RealEstate10K [12] dataset with augmented intrinsics (1-3× zoom-in) as described in Section 4.3.

estimation in Table A.4 and LVSM for novel view synthesis in Table A.3, both with augmented intrinsics in training to highlight the benefits of PRoPE on intrinsic modeling.

The new results corroborate the benefits of PRoPE that were originally shown in the main paper. We find that PRoPE consistently outperforms GTA. Importantly, both PRoPE and PRoPE+CamRay outperform GTA+CamRay, despite the fact that these methods include the same information and are all invariant to the choice of global frame. This supports our finding that attention-level intrinsic conditioning (as in PRoPE) is important.

### A.2.3 Qualitative Results for Stereo Depth Estimation

In Figure A.2 we show more qualitative results on the task of stereo depth estimation with Uni-Match [14].

| Method | 1.0× | 1.5× | 2.0× | 3.0× |
|---|---|---|---|---|
| UniMatch | 0.234 | 0.228 | 0.238 | 0.299 |
| +Plucker | -2.47% | -9.35% | -7.74% | -4.67% |
| +GTA | -28.83% | -33.48% | -32.15% | -12.55% |
| +GTA+CamRay | -29.90% | -35.25% | -32.42% | -13.65% |
| +PRoPE | **-33.75%** | **-36.66%** | **-33.85%** | -26.40% |
| +PRoPE+CamRay | -31.99% | -35.82% | -33.24% | **-28.21%** |

abs_rel ($\downarrow$) across zoom-in levels.

| Method | 1.0× | 1.5× | 2.0× | 3.0× |
|---|---|---|---|---|
| UniMatch | 0.357 | 0.376 | 0.408 | 0.613 |
| +Plucker | -1.08% | -9.46% | -2.76% | -3.37% |
| +GTA | -31.77% | -41.52% | -41.16% | -11.88% |
| +GTA+CamRay | -33.54% | -44.06% | -42.34% | -13.09% |
| +PRoPE | **-45.13%** | **-46.79%** | **-44.53%** | -38.56% |
| +PRoPE+CamRay | -41.79% | -46.75% | -44.08% | **-39.88%** |

sq_rel ($\downarrow$) across zoom-in levels.

| Method | 1.0× | 1.5× | 2.0× | 3.0× |
|---|---|---|---|---|
| UniMatch | 1.059 | 1.031 | 1.035 | 1.188 |
| +Plucker | +1.03% | -3.95% | -0.53% | +1.64% |
| +GTA | -19.55% | -26.12% | -25.68% | -4.50% |
| +GTA+CamRay | -20.41% | -27.44% | -25.72% | -6.12% |
| +PRoPE | **-28.64%** | **-29.79%** | **-28.25%** | -22.61% |
| +PRoPE+CamRay | -26.86% | -29.29% | -28.09% | **-23.74%** |

rmse ($\downarrow$) across zoom-in levels.

| Method | 1.0× | 1.5× | 2.0× | 3.0× |
|---|---|---|---|---|
| UniMatch | 0.322 | 0.322 | 0.336 | 0.421 |
| +Plucker | +0.49% | -5.46% | -2.33% | +2.00% |
| +GTA | -23.29% | -30.33% | -30.03% | -3.24% |
| +GTA+CamRay | -24.08% | -31.64% | -30.23% | -4.34% |
| +PRoPE | **-32.01%** | **-33.56%** | **-32.67%** | -26.78% |
| +PRoPE+CamRay | -29.98% | -32.68% | -32.31% | **-27.71%** |

rmse_log ($\downarrow$) across zoom-in levels.

Table A.4: **Additional Stereo Depth Estimation Results on Out-of-distribution Intrinsics at Test Time.** Experiments are conducted with UniMatch [14]'s official code as described in Section 4.6 but with 1/8 of the training resources (2 GPUs x 50k steps).

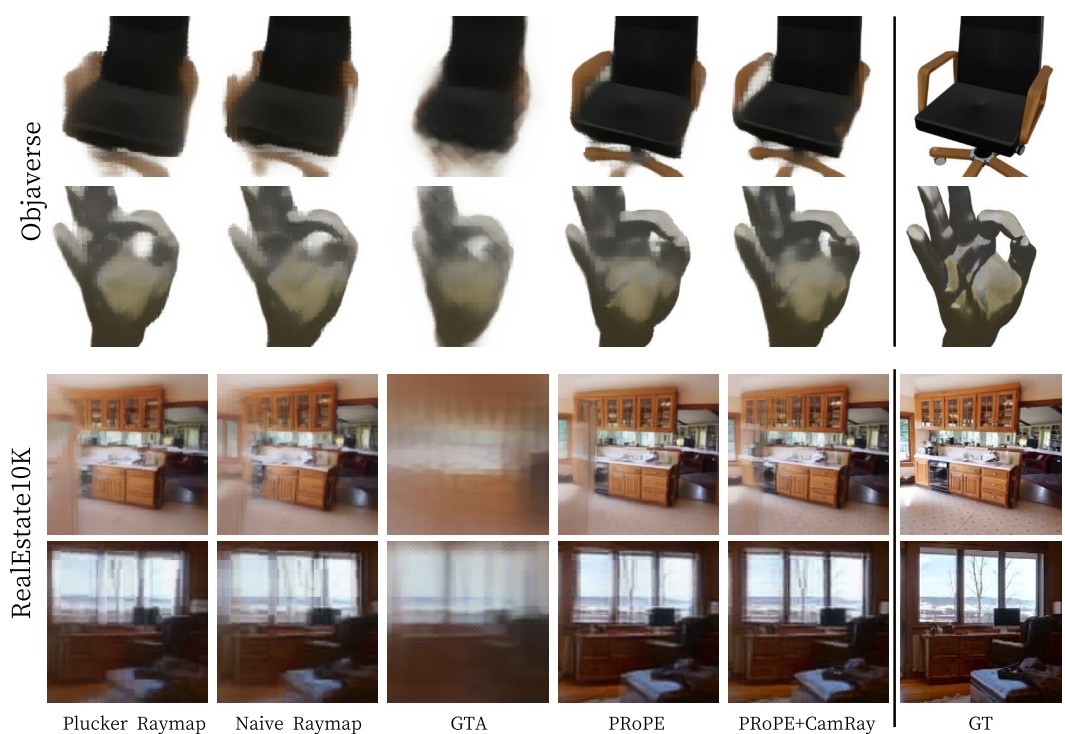

| Plucker Raymap | Naive Raymap | GTA | PRoPE | PRoPE+CamRay | GT |

Figure A.1: **More Qualitative Results of Novel View Synthesis on RealEstate10K [12] and Objaverse [13].**

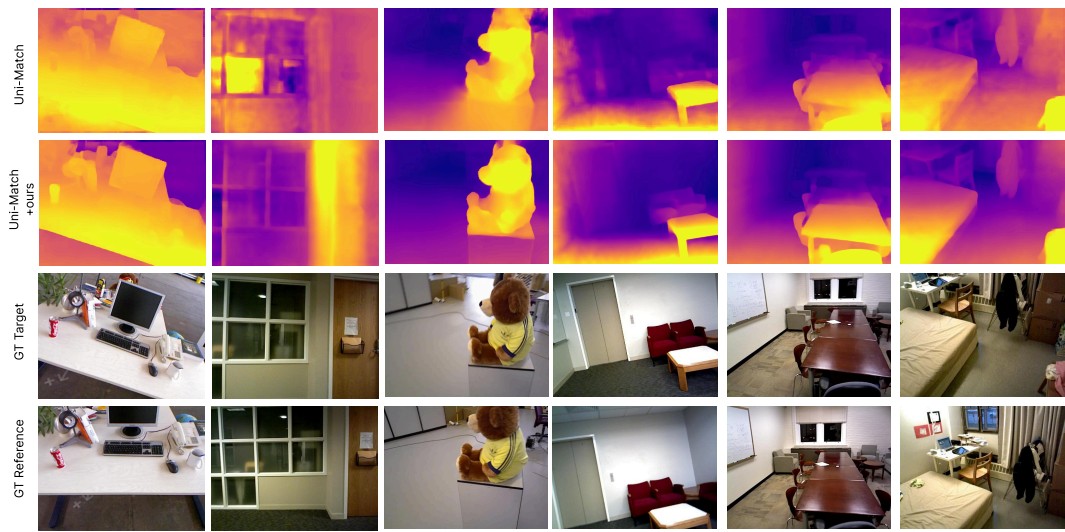

Figure A.2: **More Qualitative Results of Stereo Depth Estimation on RGBD [62], SUN3D [63] and Scenes11 [64].**

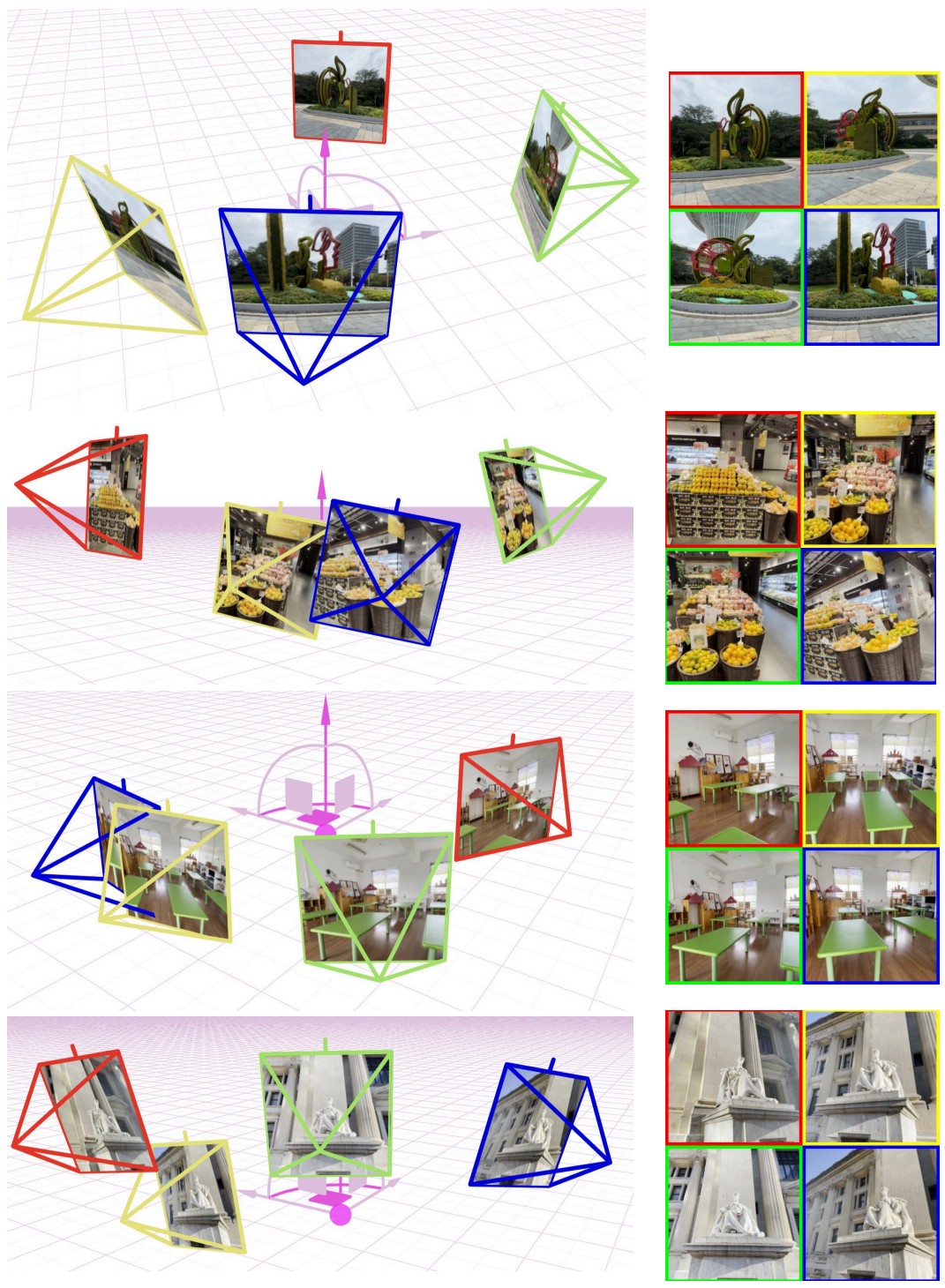

Figure A.3: **The Spatial Cognition Task.** The model takes multi-view images and corresponding camera information as input and aims to identify inconsistent image-camera pairs (yellow here). Understanding the cross-view relationships between images and cameras is crucial for this task.

