# OpenReview forum: "Cameras as Relative Positional Encoding"
_NeurIPS.cc/2025/Conference — NeurIPS 2025 poster_

### Official Review · Reviewer_BUBt · 2025-06-03

**Clarity:** 4
**Significance:** 3
**Originality:** 3
**Rating:** 5
**Confidence:** 4

**Summary:**

This paper introduces a new relative positional encoding scheme for 3D-conditioned tasks which includes the full projective relationship (intrinsics added to prior work including relative 3D encoding just with extrinsics). They demonstrate strong results on improving 3D-conditioned tasks, especially those with out-of-distribution intrinsics.

**Questions:**

- How much does including intrinsics help over just extrinsics (GTA)? Figure 3 provides good illustration but more robust and extensive comparison (including the GTA baseline in all experimental results ideally) would be better
- Is there any comparison you can add with different camera models? E.g. with fish-eye or lens-distorted models or is your method limited to just the pinhole camera?

**Ethical Concerns:**

["NO or VERY MINOR ethics concerns only"]

**Final Justification:**

My opinion has not changed after the author responses: this paper presents a small and elegant change for per-pixel relative-SE3 conditioning that now encodes intrinsics with an intuitive formulation. I believe it warrants acceptance even without the non-pinhole camera distortion modeling because non-pinhole cameras are more rare and can be handled with pre-processing or likely a simple extension to the formulation to include the distortion parameters.

**Limitations:**

- Unclear how to extend encoding multi-frequency embedding for higher-frequency prediction, but prior work also had this limitation and not the focus of this work
- Method requires known intrinsics which can be difficult to obtain (but still highly useful when one has them)

**Paper Formatting Concerns:**

No major concerns.

**Quality:**

3

**Strengths And Weaknesses:**

Strengths:

- Paper is clear and easy to follow with sound experiments and results
- Well-defined scope and contribution (adding intrinsics to the relative 3D encoding instead of just extrinsics)
- Simple implementation with few lines of code modification for strong results

Weaknesses:

- Contribution over prior work is necessary (adding intrinsics to the relative transformations) but not a very strong/novel change
- Should use additional baseline comparisons, particularly GTA which is the most similar baseline to demonstrate how important the intrinsics encoding are, e.g. in table 1 and figure 6 for the CAT3D and UniMatch comparisons

---

> ### Author Rebuttal · Authors · 2025-07-28
>
> We thank Reviewer BUBt02 for their thorough review, and recognizing strengths of the work: clarity, soundness of experiments, scope and contribution, and simple implementation.
>
> **Q: "How much does including intrinsics help over just extrinsics (GTA)?"**
>
> We ran additional experiments to quantify this, in particular by evaluating GTA+CamRay to measure the improvement from including intrinsics information. We observe that intrinsics are generally critical: GTA+CamRay outperforms GTA significantly. Importantly, we also see that PRoPE and PRoPE+CamRay *both* outperform GTA+CamRay. This shows that the method used to condition on intrinsics matters; the PRoPE-based approach for this conditioning is empirically advantageous than raymap based such as Plucker. Tables can be found below.
>
> **Q: "Is your method limited to just the pinhole camera?"**
>
> We have not yet evaluated PRoPE on distorted images; but we believe our main conclusions will still be highly relevant hold. Potential approaches for handling distorted cameras are discussed in our response to Reviewer GRaq; we view these as promising directions for follow-up work. We will update our paper with discussion.
>
>
> ---
> ## More Analysis: LVSM for Novel-View-Synthesis
>
> Here we report the complete analysis of different conditioning approaches with LVSM on Re10K dataset. The setting is same as the the experiment reported in supplemental document Table 1 (Intrinsic-aware experiment with random 1-3x zoom-in in during training) . Here we additionally report results with GTA+CamRay, as well as the evaluation on OOD intrinsics with 1x -7x zoom-in in at test time.
>
> **Analysis**
> - GTA v.s. GTA+CamRay: GTA itself does not encode camera intrinsics. While the network can implicit learn the intrinsic from pixels, it is much less effective than explicitly providing the intrinsic to the model.
> - Plücker v.s. GTA+CamRay: Both of them provide {extrinsic, intrinsic} to the model. GTA+CamRay consistently outperform Plücker by ~2db on PSNR indicates attention-level conditioning is superior than pixel-level conditioning.
> - GTA v.s. PRoPE: A simple change of introducing intrinsic into the formula (one line change) boost performance by 4db-6db on PSNR.
> - GTA+CamRay v.s. PRoPE. As camera intrinsics gradually becomes OOD, the margin between attention level conditioning (PRoPE)  and pixel-aligned conditioning (GTA+CamRay) increases for 0db to 0.7db. This indicates attention level conditioning is more robust to OOD data.
> - PRoPE v.s. PRoPE+CamRay. On top of attention-level conditioning, which is applied per-token, CamRay additionally provides pixel-level conditioning thus could further help with the performance. And the fact that it only affect performance marginally indicates the effectiveness of PRoPE on encoding intrinsics.
>
> **Metric: PSNR** (higher is better)
>
> | ZoomIn | Plücker | GTA    | GTA+CamRay | PRoPE  | PRoPE+CamRay |
> | ------ | ------- | ------ | ---------- | ------ | ------------ |
> | 1.0x   | 19.889  | 15.772 | 21.414     | 21.419 | 21.856       |
> | 3.0x   | 21.031  | 18.142 | 22.703     | 22.951 | 23.061       |
> | 5.0x   | 20.749  | 18.123 | 22.345     | 22.748 | 22.833       |
> | 7.0x   | 20.322  | 18.286 | 21.772     | 22.400 | 22.401       |
>
>
> **Metric: SSIM** (higher is better)
>
> | ZoomIn | Plücker | GTA   | GTA+CamRay | PRoPE | PRoPE+CamRay |
> | ------ | ------- | ----- | ---------- | ----- | ------------ |
> | 1.0x   | 0.608   | 0.512 | 0.673      | 0.678 | 0.693        |
> | 3.0x   | 0.694   | 0.626 | 0.741      | 0.748 | 0.751        |
> | 5.0x   | 0.732   | 0.683 | 0.764      | 0.775 | 0.776        |
> | 7.0x   | 0.755   | 0.721 | 0.780      | 0.794 | 0.794        |
>
>
> **Metric: LPIPS** (lower is better)
>
> | ZoomIn | Plücker | GTA   | GTA+CamRay | PRoPE | PRoPE+CamRay |
> | ------ | ------- | ----- | ---------- | ----- | ------------ |
> | 1.0x   | 0.327   | 0.641 | 0.238      | 0.247 | 0.218        |
> | 3.0x   | 0.272   | 0.488 | 0.191      | 0.193 | 0.183        |
> | 5.0x   | 0.308   | 0.439 | 0.242      | 0.227 | 0.220        |
> | 7.0x   | 0.345   | 0.403 | 0.288      | 0.244 | 0.242        |
>
> ---
> ## More Analysis: UniMatch for Stereo Depth Estimation.
>
> Here we report the complete analysis of different conditioning approaches with UniMatch for stereo depth estimation. The setting is same as the the experiment reported in main paper Table 2, but due to resource constrain, these models are trained with 1/8 of the resources (2 GPUs x 50k steps) comparing to what we reported in the paper (8 GPUs x 100k steps). Here we report the **average relative error reduction (more negative is better)** across three test sets—RGBD, SUN3D, and Scenes11—under test-time zoom-in of {1.0x, 1.5x, 2.0x, 3.0x} to simulate cameras with OOD intrinsics.
>
> **Analysis**. Across all metrics and zoom-in levels, PRoPE consistently outperforms GTA and GTA+CamRay with noticeable margins, highlighting the effectiveness of encoding camera intrinsics directly at the attention level. Notably, under the most extreme OOD condition (3.0x zoom), PRoPE continues to yield substantial gains (22–38%), whereas GTA+CamRay’s improvements drop to just 4–13%. This demonstrates the superior generalization and robustness of PRoPE's design in challenging OOD scenarios. When using CamRay with GTA, it consistently improves the performance by 1%-4%. We note that CamRay does not always bring benefits to PRoPE: this indicates that PRoPE alone is already conditioning effectively on intrinsics.
>
>
>
> **Metric Improvement: abs\_rel** (more negative is better)
>
> | ZoomIn | UniMatch | +Plucker | +GTA    | +GTA+CamRay | +PRoPE  | +PRoPE+CamRay |
> | ------ | -------- | -------- | ------- | ----------- | ------- | ------------- |
> | 1.0x   | 0.234133 | -2.47%   | -28.83% | -29.90%     | -33.75% | -31.99%       |
> | 1.5x   | 0.228133 | -9.35%   | -33.48% | -35.25%     | -36.66% | -35.82%       |
> | 2.0x   | 0.238433 | -7.74%   | -32.15% | -32.42%     | -33.85% | -33.24%       |
> | 3.0x   | 0.298500 | -4.67%   | -12.55% | -13.65%     | -26.40% | -28.21%       |
>
>
> **Metric Improvement: sq\_rel** (more negative is better)
>
> | ZoomIn | UniMatch | +Plucker | +GTA    | +GTA+CamRay | +PRoPE  | +PRoPE+CamRay |
> | ------ | -------- | -------- | ------- | ----------- | ------- | ------------- |
> | 1.0x   | 0.357467 | -1.08%   | -31.77% | -33.54%     | -45.13% | -41.79%       |
> | 1.5x   | 0.376133 | -9.46%   | -41.52% | -44.06%     | -46.79% | -46.75%       |
> | 2.0x   | 0.407833 | -2.76%   | -41.16% | -42.34%     | -44.53% | -44.08%       |
> | 3.0x   | 0.612533 | -3.37%   | -11.88% | -13.09%     | -38.56% | -39.88%       |
>
>
> **Metric Improvement: rmse** (more negative is better)
>
> | ZoomIn | UniMatch | +Plucker | +GTA    | +GTA+CamRay | +PRoPE  | +PRoPE+CamRay |
> | ------ | -------- | -------- | ------- | ----------- | ------- | ------------- |
> | 1.0x   | 1.058767 | 1.03%    | -19.55% | -20.41%     | -28.64% | -26.86%       |
> | 1.5x   | 1.030967 | -3.95%   | -26.12% | -27.44%     | -29.79% | -29.29%       |
> | 2.0x   | 1.034633 | -0.53%   | -25.68% | -25.72%     | -28.25% | -28.09%       |
> | 3.0x   | 1.188033 | 1.64%    | -4.50%  | -6.12%      | -22.61% | -23.74%       |
>
> **Metric Improvement: rmse\_log** (more negative is better)
>
> | ZoomIn | UniMatch | +Plucker | +GTA    | +GTA+CamRay | +PRoPE  | +PRoPE+CamRay |
> | ------ | -------- | -------- | ------- | ----------- | ------- | ------------- |
> | 1.0x   | 0.321733 | 0.49%    | -23.29% | -24.08%     | -32.01% | -29.98%       |
> | 1.5x   | 0.321867 | -5.46%   | -30.33% | -31.64%     | -33.56% | -32.68%       |
> | 2.0x   | 0.335533 | -2.33%   | -30.03% | -30.23%     | -32.67% | -32.31%       |
> | 3.0x   | 0.420733 | 2.00%    | -3.24%  | -4.34%      | -26.78% | -27.71%       |

---

> > ### Comment · Reviewer_BUBt · 2025-08-04
> >
> > Thank you for re-emphasizing the difference on using intrinsics vs not and for clarifying the future direction of modeling non-pinhole intrinsics. I have no additional questions.

---

> > > ### Author Response · Authors · 2025-08-06
> > >
> > > Thank you again for your thoughtful review. We're pleased that your concerns have been clarified.

---

### Official Review · Reviewer_19jr · 2025-06-27

**Clarity:** 3
**Significance:** 2
**Originality:** 2
**Rating:** 4
**Confidence:** 5

**Summary:**

This paper introduces PRoPE, a projective positional encoding for multi-view transformers that unifies intrinsic and extrinsic camera parameters into a global-frame-invariant, attention-level representation. The method extends prior SE(3)-based encodings by modeling full projective transformations between views. A complementary token-level encoding, CamRay, is used to provide dense per-pixel intrinsics in the local frame. Experiments on novel view synthesis, stereo depth estimation, and spatial consistency show that PRoPE improves generalization, particularly when test-time focal lengths or number of input views differ from training.

**Questions:**

- Would GTA+CamRay or CAPE+CamRay close the gap to PRoPE? This comparison is critical for fair evaluation, as it ensures that intrinsic information is made available to the baseline models. Without this, it is difficult to attribute performance gains to PRoPE’s projective formulation rather than the presence of intrinsics. This is especially important in scenarios with varying focal lengths, where intrinsic-aware baselines may perform comparably if given the same input.

- Can you ablate Q/K vs. Q/K/V/O injection? GTA uses all Q/K/V/O vectors, while methods like RoPE and CAPE apply transformations only to Q/K. It would be valuable to understand whether PRoPE requires the full injection or if a simpler variant would suffice.

- Have you tested PRoPE under more challenging camera models, such as fisheye lenses or other camera model? These cases introduce nonlinear or non-projective distortions, and it is unclear whether PRoPE generalizes to them.

- How numerically stable is PRoPE when projection matrices are ill-conditioned or when intrinsic parameters vary significantly? For example, extremely small focal lengths or near-degenerate calibration settings may cause instability—how does PRoPE behave in such regimes?

**Ethical Concerns:**

["NO or VERY MINOR ethics concerns only"]

**Final Justification:**

After the rebuttal and the discussion with the authors, I believe the additional results and clarification resolved most of my concerns and greatly strengthened the paper. The improvement over baseline is still marginal, but I would like to upgrade my score accordingly.

**Limitations:**

The paper mentions numerical instability and lack of multi-frequency modeling as future directions.

**Quality:**

3

**Strengths And Weaknesses:**

Strengths

- PRoPE cleanly generalizes attention-level geometric encodings by incorporating intrinsic parameters in a projectively correct and invariant manner.

- The encoding is compatible with FlashAttention and standard transformer layers, and requires minimal architectural changes.

- The model performs well under distribution shifts (e.g., varying focal length or number of views), which are realistic but often overlooked evaluation settings.

Weakness


- While mathematically sound, PRoPE is fundamentally an extension of existing methods like CAPE and GTA, augmented with intrinsics. Since intrinsics are low-dimensional and already used in token-level encodings, the conceptual contribution is relatively modest.

- Crucially, the paper does not evaluate GTA+CamRay or CAPE+CamRay, which are the most relevant and fair baselines for comparison. These combinations would test whether the gains from PRoPE stem from its projective formulation or simply from the addition of CamRay-like intrinsic information. Their absence prevents a meaningful comparison and undermines the empirical evidence for PRoPE’s advantage.

- The choice to inject projective encoding into all Q/K/V/O vectors (following GTA) is not ablated or justified against simpler Q/K-only alternatives in RoPE.

- The method is named as PRoPE, and frequency modeling is an important desgin in RoPE. A more interesting and fundamental direction is to consider the frequency modeling in SE3 and 3D geometric space.

- The key contribution of this paper is extending intrinsic encoding to prior SE3 encodings, but how generalise the intrinsic encoding is for different camera models and extreme focus length is not fully analyzed.

---

> ### Author Rebuttal · Authors · 2025-07-28
>
> We thank Reviewer 19jr for their comprehensive and constructive feedback. We include responses to key points below, which we will use to strengthen our paper.
>
> **"Would GTA+CamRay or CAPE+CamRay close the gap to PRoPE?"**
>
> This is an excellent question. We have run a more exhaustive set of experiments to answer it; to reduce redundancy, tables can be found in our response Reviewer BUBt.
>
> Importantly, we observe that PRoPE and PRoPE+CamRay outperform GTA+CamRay for both novel view synthesis and for stereo depth estimation, despite the fact that these methods include the same information and invariance properties. This corroborates our finding that PRoPE's attention-level intrinsic conditioning is important. We will include these results in our updated paper.
>
> **"Can you ablate Q/K vs. Q/K/V/O injection?"**
>
> We have run additional experiments ablating this decision on LVSM with the Re10k dataset (with intrinsics augmentation during training). Overall, they are consistent with the results in the original GTA paper: Q/K/V/O injection produces small but consistent improvement in results..
>
> **Metric: PSNR** (higher is better)
>
> | Transform | GTA    | GTA+CamRay | PRoPE  | PRoPE+CamRay |
> | --------- | ------ | ---------- | ------ | ------------ |
> | QK        | 15.958 | 21.332     | 21.350 | 21.688       |
> | QKVO      | 15.772 | 21.414     | 21.419 | 21.856       |
>
> **Metric: SSIM** (higher is better)
>
> | Transform | GTA    | GTA+CamRay | PRoPE  | PRoPE+CamRay |
> | --------- | ------ | ---------- | ------ | ------------ |
> | QK        | 0.513  | 0.670      | 0.672  | 0.685        |
> | QKVO      | 0.512  | 0.673      | 0.678  | 0.693        |
>
> **Metric: LPIPS** (lower is better)
>
> | Transform | GTA    | GTA+CamRay | PRoPE  | PRoPE+CamRay |
> | --------- | ------ | ---------- | ------ | ------------ |
> | QK        |  0.639 | 0.250      | 0.269  | 0.231        |
> | QKVO      |  0.641 | 0.238      | 0.247  | 0.218        |
>
> **Q: "Have you tested PRoPE under more challenging camera models, such as fisheye lenses or other camera model?", "How numerically stable is PRoPE when projection matrices are ill-conditioned or when intrinsic parameters vary significantly?"**
>
> We have not yet evaluated PRoPE for heavily distorted images, but we believe the key conclusions in our paper still apply to these cases. We discuss potential approaches for handling distorted cameras in our response to Reviewer GRaq. This would be an exciting direction for followup work. Degenerate cases: we have not observed any instability associated with ill-conditioned projection matrices, but we agree that this is a possible in adversarial data. We will emphasize it more heavily in our future work section. We have not observed any instability associated with significantly varying intrinsics, which we do evaluate it via OOD intrinsics augmentation (up to 7x zoom-in).
>
> **"Conceptual contribution is relatively modest"**
>
> We agree with the reviewer that PRoPE builds heavily on existing ideas in relative positional encoding. We also believe, however, that the practical significance is still large. Plücker raymaps remain the dominant approach for conditioning multiview transformers, primarily because they unify representation for both camera intrinsics and extrinsics. Our work aims to show that there are better ways to implement these models, which is increasingly important as they are scaled. We also believe that the technical survey and comparative analysis in the paper will be a valuable resource for the 3D computer vision community.

---

> > ### Comment · Reviewer_19jr · 2025-08-04
> > **GTA/CaPE + CamRay as the essential baselines**
> >
> > I appreciate the authors' response and the additional results provided here and in the discussion with Reviewer BUBt.
> >
> > As mentioned in my original review and echoed by other reviewers, **a key concern remains:
> > Would GTA+CamRay or CAPE+CamRay close the gap to PRoPE?**
> > This comparison is critical for fair evaluation, as it ensures that **intrinsic information is equally available to all methods**. Without it, it is difficult to attribute PRoPE’s gains to its projective formulation rather than to the inclusion of intrinsic cues.
> >
> > The newly reported results show that **baselines with CamRay consistently perform slightly worse than PRoPE, but the differences are often marginal, sometimes within 0.0x PSNR**. This raises further concern about the novelty and practical significance of the proposed method. If prior methods combined with CamRay already achieve comparable performance, it suggests that PRoPE may offer limited modeling improvement beyond enabling access to intrinsics.
> >
> > Importantly, these findings **should be included in the main paper and faithfully discussed**, rather than only appearing in the rebuttal. Without this transparency, the empirical support for PRoPE's claimed advantage remains incomplete.
> >
> > Overall, while **I appreciate the effort to improve pose encoding in multi-view transformers**, the theoretical contribution remains modest and the observed performance gain appears incremental. A more complete and balanced presentation would strengthen the work.

---

> ### Author Response · Authors · 2025-08-04
>
> Dear Reviewer,
>
> We would like to thank you for your thoughtful follow-up.
>
> For you concern on **Would GTA+CamRay or CAPE+CamRay close the gap to PRoPE?**, we would like to kindly point out the following observations from our experiments:
>
> 1. **PRoPE is much more robust than GTA+CamRay to OOD intrinsics**. In both the LVSM and UniMatch experiments, the gap between the two gets larger and larger when the test data contains more and more OOD intrinsics:
> - LVSM: From 1x -> 7x, PSNR gap is 0.0 -> 0.7; LPIPS gap is -0.009 -> 0.044
> - UniMatch: From 1x -> 3x, RMSE improvement gap is 8% -> 16% (similar for all other metrics)
>
> 2. **PRoPE+CamRay consistently outperforms GTA+CamRay in all experiments** : This suggests PRoPE + CamRay should be the go-to choice over GTA+CamRay.
>
> 2. **GTA+CamRay itself is novel -- An outcome of our thorough analysis**. The thorough analysis of different approaches is our major contribution, which leads to the both the GTA+CamRay and PRoPE. It is worth to note that none of the existing works realize this solution even though it is simple. We believe our analysis [not only PRoPE] would bring valuable insights to the community.
>
> We appreciate reviewer again for the valuable feedbacks. These results and analysis will be added into the revised version.

---

> > ### Comment · Reviewer_19jr · 2025-08-05
> > **Highlight the intrinsic effect**
> >
> > Thanks for authors further clarification.
> >
> > The improvement of PRoPE over focal length change makes sense, and it highlights the contribution. I appreciate it.
> >
> > As the results show the focal length change from 1x -> 7x, I'm curious how the effect will be if change from 1x -> 0.1x, to the other way around.

---

> > > ### Author Response · Authors · 2025-08-05
> > >
> > > Thank you for the follow-up question — it's a great one, and we’ve considered running such an experiment as well. Unfortunately, it's not practical in the current setup because all of these method (LVSM, UniMatch etc) rely on both the RGB images as input and ground-truth for metric evaluation. While it's feasible to simulate zoom-in scenarios through cropping and rescaling, generating zoomed-out images is not straightforward, as it would require information from regions outside the original image, which we simply don't have.
> > >
> > > Still, we appreciate your thoughtful suggestion!

---

> > > > ### Comment · Reviewer_19jr · 2025-08-07
> > > >
> > > > I appreciate authors' further clarification. That makes sense. After the discussion and the additional results, I believe the submission is much stronger and valid. I would like to raise my score accordingly.

---

> > > > > ### Author Response · Authors · 2025-08-08
> > > > >
> > > > > Thank you for the constructive discussion. Your suggestions have been invaluable, and we truly appreciate the time and effort you dedicated to reviewing our work. We are glad to hear that our discussion has helped clarify the value of our contribution and that you are now considering a higher score. We sincerely appreciate your willingness to revisit your evaluation.

---

### Official Review · Reviewer_GRaq · 2025-07-01

**Clarity:** 3
**Significance:** 2
**Originality:** 1
**Rating:** 4
**Confidence:** 4

**Summary:**

The paper proposes a new positional encoding for multi-view transformers applied to 3D perception tasks. In this context, encoding and injecting 3D information inside the transformer architecture grounds the model in 3D and is critical for the quality of the resulting predictions. Existing approaches inject these priors either by concatenating dense raymaps to input images before the tokenization or by directly injecting relative pose information in the attention layers. Raymaps are not invariant with regard to transformations between global coordinate systems while existing methods encoding relative pose do not take into account intrinsic information which may hinder performances on testing views with different intrinsics. The authors therefore introduce PRoPE which encodes pairwise projective transformations and injects it in self attention layers. In addition, they introduce CamRays which express dense camera rays in the camera coordinate space instead of the world coordinate space.
Applied to models trained for novel view synthesis and depth prediction tasks, PRoPE and CamRays increase the performances of such 3D multi-view transformers and their ability to generalize to different testing intrinsics.

**Questions:**

Limited contribution is not necessarily a restricting factor, however I feel like there should be a more comprehensive set of experiments to clearly demonstrate the benefits of your method, pinpoint the limitations and highlight the scenarios in which your encodings are the most beneficial. For example I find the Intrinsic-aware Benchmarking experiment from the supplementary material to be more relevant than Tab.1 and Tab.2.

In the spatial cognition task and setting 1 of RealEstate10k experiments , what is the range of viewpoint change between cameras? Can you quantify how CamRay performs vs Plucker embeddings on different magnitude of viewpoint changes?

As an alternative, have you considered encoding epipolar information between pairs of images? Intrinsics information could also be included in the fundamental matrix.

Could your model be easily extended to account for distortion?

**Ethical Concerns:**

["NO or VERY MINOR ethics concerns only"]

**Final Justification:**

The additional experiments show that ProPE consistently outperforms GTA across tasks and better highlights the benefits of directly integrating intrinsic information in the attention layers. The paper heavily relies on GTA and provide only intuitive incremental contributions. However, supported by a comprehensive experimental section, these incremental contributions hold value and do not justify a rejection.
Following the rebutall and discussions I have increased my score accordingly.

**Limitations:**

Yes

**Quality:**

2

**Strengths And Weaknesses:**

### Strengths
1) The problems associated with existing encoding approaches are well identified. The proposed solutions are sound and tackle the identified shortcomings in an effective way as demonstrated in the experimental section. Injecting intrinsic information explicitly provides camera prior which is more effective than letting the network implicitly capture intrinsic information.

2) Similarly, CamRay consistently outperforms Plucker rays embedding by preserving invariance wrt. the global frame which allows for better generalization to a larger number of viewpoints demonstrated in the experimental section.

3) The spatial cognition task is an interesting additional experiment that goes beyond existing evaluation metrics and seems to be a good proxy for evaluating the geometric scene understanding of the model.

### Weaknesses
1) ProPE is very largely inspired by GTA which limits the contributions of the paper. As RoPE is equivalent to SO(2) elements, the only notable difference with GTA is the introduction of intrinsic matrices when computing the relative transformation between pairs of cameras.
It has to be noted that the authors clearly stated that ProPE is an extension of existing work and formulated the contributions in that direction.

2) This leads to my second concern. Tab. 1 and Tab. 2 show that applying PRoPE to multi-view diffusion models or stereo depth estimation models improve their performances. However as PRoPE is an extension of GTA, if the intrinsic variations within the dataset are limited/non-existent PRoPE is not expected to perform better than GTA ( see Fig. 3 a, GTA and PRoPE seem to behave similarly). For the same reason GTA is also expected to improve the models performance on these tasks.
I think it would be less disingenuous to also report CAT3D+GTA and UniMatch+GTA. And if PRoPE improves upon GTA in these experiments, use it to highlight the benefit of your method.

3) As stated by the authors, the main motivation for encoding intrinsic information is application to real world scenarios. However real world cameras have camera models with distortion, as such using this method would require an additional undistortion step in most cases. It would be an interesting addition to also encode and inject distortion parameters in the models.

4) I find the Figures related to the Spatial Cognition task to be more confusing than informative, it is not obvious to identify the mismatched pair at a glance.

5) Fig 3 b), the x labels {1,3,5} zoom factors on the graphs do not match with the {1,2,3} zoom factors in the legend.

---

> ### Author Rebuttal · Authors · 2025-07-28
>
> We thank Reviewer GRaq for the thoroughness of their review, including: recognizing the soundness of proposed solutions, ability to retain invariance with CamRay, and value of the spatial cognition experiment.
>
> **Q: "Also report CAT3D+GTA and UniMatch+GTA", "more comprehensive set of experiments to clearly demonstrate the benefits of [PRoPE]".**
>
> We thank the reviewer for these suggestions: we agree that more comprehensive experiments would strengthen our conclusions. We have therefore run more experiments using a more thorough combination of camera conditioning techniques on both UniMatch for stereo depth estimation and LVSM for novel view synthesis, both with augmented intrinsics in training to highlight the difference benefits of PRoPE. To minimize redundancy, we include these results in our response to Reviewer BUBt.
>
> The new results corroborate the benefits of PRoPE that were originally shown in Figure 3b and Supplemental Table 1 in our submission. We find that PRoPE consistently outperforms GTA. Importantly, both PRoPE and PRoPE+CamRay outperform GTA+CamRay, despite the fact that these methods include the same information and are all invariant to the choice of global frame. This supports our finding that attention-level intrinsic conditioning (as in PRoPE) is important.
>
> While there is not enough time to re-run the CAT3D experiments within the rebuttal period (as we don't have access to its code and need assistants from its authors), we hope that these additional results help alleviate your valid concerns. We will include both the results and additional analysis in the updated version of our manuscript.
>
> **Q: "Could your model be easily extended to account for distortion?"**
>
> This is an excellent point; we will update our paper with discussion about this limitation and possible future directions. We believe that our high-level conclusion that relative information is important still applies. As the reviewer notes, the most direct route for incorporating distortion into our method is to undistort images before passing them into the model. Other options include injecting projective approximations (e.g., via linearization) into the attention-level encoding (e.g., each token/patch have a different intrinsic K), while relying on CamRay for more fine-grained distortion handling. It would be interesting to explore these methods for datasets with known distortion parameters. We consider this out of scope for our work, but agree that it is an exciting area for future research.
>
> **Q: "Can you quantify [...] on different magnitude of viewpoint changes?"**
>
> We use the standard test sets in each task which contains both small or large viewpoint changes. However it's challenging to quantify based on viewpoint magnitude although that is a good suggestion because both rotation, translation, and perspective changes need to be considered. And to our best knowledge there is no standard way of doing so.
>
> **Q: "Have you considered encoding epipolar information between pairs of images?"**
>
> Good question. In our initial experiments, we explored encoding the relative geometric distance between ray pairs (from corresponding pixels) in 3D space using the fundamental matrix. Under this approach, rays lying on the same epipolar plane would have zero relative distance. However, we found this encoding to be fundamentally ambiguous -- multiple distinct ray configurations can yield identical relative distance values. This limitation means that epipolar constraints alone cannot capture the complete relative geometric information needed for effective intrinsic encoding, resulting in undesired performance.

---

> > ### Comment · Reviewer_GRaq · 2025-08-05
> >
> > Thank you for the rebuttal and for providing useful additional experimental results. I have no further questions at this moment.

---

> > > ### Author Response · Authors · 2025-08-06
> > >
> > > Thank you for your thoughtful response and for taking the time to review our paper. We're glad to hear that your concerns have been addressed. If any further questions arise, please don’t hesitate to reach out—we’d be happy to continue the discussion.
> > >
> > > If our response has sufficiently addressed your main concerns, we would be grateful if you would consider revisiting your score. We truly appreciate your time and effort in helping to improve our work.

---

### Official Review · Reviewer_X9c5 · 2025-07-03

**Clarity:** 3
**Significance:** 4
**Originality:** 3
**Rating:** 5
**Confidence:** 4

**Summary:**

## Context
Recently, multi-view transformer architectures became increasingly popular for 3D tasks. For each input view, the viewpoint, in terms of camera extrinsics and intrinsics, needs to be provided to the model. Previous works use different ways to provide those camera parameters to the model, summarized by the paper under review as follows:
- token-level conditioning:
	- concatenate camera conditioning to the respective image tokens
	- methods: raymaps, Plücker raymaps
	- can encode extrinsics and intrinsics
	- encoding depends on the choice of the reference frame
	- sensitive to global SE(3) transformations
- attention-level conditioning:
	- inject pairwise camera conditioning into the attention mechanism
	- methods: CAPE (from EscherNet), GTA
	- existing methods only excode extrinsics, but not intrinsics
	- encode relative poses between pairs of views and are therefore invariant to the reference frame
	- invariant to global SE(3) transformations

## Method
To resolve the shortcomings of previous methods, the paper under review proposes a new encoding method called PRoPE+CamRay. PRoPE essentially extends GTA to also incorporate the intrinsics into the relative positional encoding between pairwise views. CamRay additionally provides the intrinsics to the model via token-level conditioning in form of an intrinsics-only-raymap.

## Experiments
The most relevant experiments are conducted for the task of novel view synthesis on RealEstate10k, using LVSM as base model. The results (Fig. 3, 4, 5) show that PRoPE very slightly outperforms the best previous method GTA regarding a) performance with similar settings at test time as in training, and b) more input views at test time. In case of c) using different focal lengths at test time, it clearly outperforms GTA. The PRoPE+CamRay variant always outperforms PRoPE and has a larger margin to the best previous method GTA.

Further experiments are conducted with CAT3D, with LVSM on Objaverse, for stereo depth estimation with UniMatch on 3 datasets, and for a spatial cognition-task on DL3DV. The comparisons in those experiments are less conclusive (not all conditioning methods are evaluated). However, in all results, using PRoPE outperforms the performance of the respective baseline model.

**Questions:**

My key suggestions are:
- Provide an intuitive explanation why CamRay is necessary.
- Evaluate CamRay also in the UniMatch and CAT3D experiments (Tab. 1 and 2).
- Evaluate previous methods (Plücker, GTA, CAPE) also in the other tasks to make the "systematic analysis" more extensive.

**Ethical Concerns:**

["NO or VERY MINOR ethics concerns only"]

**Final Justification:**

The rebuttal adresses my questions well and I keep my "Accept" rating.

In particular, I do not share the criticism of reviewer 19jr. I agree that the proposed method builds on the existing method GTA. However, it extends it to also handle camera intrinsics, which is important in practice and is shown to improve results in the evaluations.

The evaluations that were provided in the rebuttal show that GTA+CamRay also performs well. However:
1. As I understand it, original GTA does not use CamRay and I would see this combination rather as a contribution of the paper under review, and not as a reason to reject it.
2. PRoPe and PRoPe+CamRay consistently and sometimes clearly outperform GTA+CamRay.

**Limitations:**

yes

**Paper Formatting Concerns:**

No concerns.

**Quality:**

4

**Strengths And Weaknesses:**

## Strengths

### Elegant method
The method is elegant, especially regarding how it comprises previous methods: in case of identity intrinsic matrices, PRoPE falls back to GTA/CAPE and in case of single-view inputs, PRoPE falls back to RoPE.

### Writing
The paper is very clearly written and good to follow.

### Consistent improvements
The paper under review presents results for a variety of tasks. In all cases, PRoPE improves the results. The more detailed experiments for NVS on Re10k (Fig. 3) show that PRoPE consistently outperforms the previous best GTA. Even though the improvement is small in some cases, it is consistent, and for the case of generalization to different intrinsics at test time, the improvement is significant. The additionally proposed CamRay method further boosts results.

## Weaknesses
### Unclear intuition for CamRay
The main difference of PRoPE over GTA is to incorporate the intrinsics into the relative positional encoding. However, CamRay additionally provides the intrinsics to the model via token-level conditioning. It is shown that this clearly improves the results. However, it remains unclear why this additional usage of token-level conditioning is required. Additional explanations would be helpful.

### Limited evaluation of CamRay
It seems like CamRay was only evaluated in the experiments for NVS on Re10k (Fig. 3) and the spatial cognition experiments (Tab. 3). The experiments with CAT3D (Tab. 1) and UniMatch (Tab. 2) seem to be without CamRay. It would be helpful to also evaluate CamRay in these experimental setups, to provide further evidence of its effectiveness.

### Limited systematic analysis
A major claim of the paper is a "systematic analysis" of different camera conditionings (line 4-5). However, such a systematic analysis is actually only conducted in one experimental setup, namely NVS on Re10k with LVSM (Fig. 3). The "systematic analysis" would be stronger, if it would be conducted also on the other tasks.

## Comments
Sometimes the authors write Plücker and sometimes Plucker coordinates.

In the experiments for NVS with varying focal lengths (line 214): does this consider context views with varying focal length, or rendering target views at different target focal lengths, or both?

Figures are sometimes not referenced from the text, for example Fig. 1 or Fig. 2.

Fig. 1b): why should PRoPE generalize better to unseen extrinsics? I understand the point that invariance to the reference frame helps generalization in that PRoPE should be able to generalize to unseen absolute poses, as long as the relative poses have been seen during training. However, I do not understand why it should generalize better to unseen relative pose configurations. To illustrate what I mean, here is an LLM example:
- The training dataset only contains sentences where the words "the" and "likes" appear at absolute positions 0 and 2.
- During inference, when using APE, a model would struggle if those words appear at positions 10 and 12. When using RPE, this should not be a problem, as the positional encoding is invariant to the absolute position.
- However, when those words appear at positions 0 and 12, I would expect that both APE and RPE have the same difficulties, as also this relative position is unseen.
- Actually the text describes this better than Fig. 1b) as it says that some methods are "sensitive to SE(3) transformations" (e.g. line 90), which seems to be a better description than the "Robust to unseen extrinsics" in Fig. 1.

---

> ### Author Rebuttal · Authors · 2025-07-28
>
> We thank the reviewer for recognizing the strengths of our work in terms of method, writing, and the consistency of performance improvements. We also appreciate the constructive feedback, which we respond to below.
>
> **Q: "Unclear intuition for CamRay. [...] Additional explanations would be helpful."**
>
> We will update our paper to include more discussion on CamRay.
>
> The main takeaway of CamRay is that token-level and attention-level camera encodings can be complementary. While PRoPE incorporates camera intrinsics through relative positional encoding at the attention-level, it treats each patch (token) as a unit and does not explicitly distinguish between individual pixels within a token. CamRay, on the other hand, enriches input patches with pixel-wise ray information. This adds another layer of positional detail on top of what PRoPE offers, which we empirically find improve results.
>
> We will also include more analysis, which we discuss next.
>
> **Q: "Limited evaluation of CamRay", "Limited systematic analysis"**
>
> We agree that a more exhaustive set of experiments would strengthen our paper. We have run more experiments, which include: more combinations of camera conditioning techniques, for both in-distribution and out-of-distribution data, on both the NVS and depth estimation tasks. Unfortunately, time constraints prevent us from sharing results for the CAT3D or spatial cognition tasks in time for the rebuttal deadline.
>
> Result tables can be found in our response to Reviewer BUBt. We will update our paper with these new results, which corroborate the key takeaways from our initial submission: relative encoding is critical for performance, and including intrinsics information via PRoPE provides further gains. We also find that CamRay is useful as a tool for analysis. GTA+CamRay and PRoPE+CamRay both contain the same information and are both invariant to the world coordinate system. However, PRoPE+CamRay consistently results in better model performance. This can be interpreted as an ablation that validates the usefulness of PRoPE's intrinsics-aware relative encoding.
>
> **Q: Do varying focal length experiments "consider context views with varying focal length, or rendering target views at different target focal lengths, or both?"**
>
> Both context and target views have augmented focal lengths. We will clarify this in the writing.
>
> **Q: "Why [relative encodings] should generalize better to unseen relative pose configurations"?**
>
> Thank you for pointing this out. We agree with your statement: "PRoPE should be able to generalize to unseen absolute poses, as long as the relative poses have been seen during training". The experiments we conducted would be better described as testing "generalization to unseen number of extrinsics/cameras" (the analogy to RoPE generalizes better to longer sequence in LLMs). This affects the batch-level statistics* of the extrinsics but not the extrinsics themselves. We will revisit Figure 1b and our writing to make sure it is more precise.
>
> *Consider: for each camera, the distance to the closest other camera. With the same scene but more cameras, this distance will usually decrease.

---

> > ### Comment · Reviewer_X9c5 · 2025-08-04
> >
> > Nothing to discuss from my side. Thanks for the clarifications and additional evaluations.

---

> > > ### Author Response · Authors · 2025-08-06
> > >
> > > We’re grateful for the time you spent reviewing and discussing our paper. It’s great to hear that your questions have been addressed.

---

### Comment · Area_Chair_nvMd · 2025-08-04

Dear Reviewers,

Please take a look at the rebuttal and check if your questions and concerns have been addressed and everything is clear. Now is the time to clarify any remaining issues in discussion with the authors.

Thanks,
Your AC

---

### Decision · Program_Chairs · 2025-09-17

**Decision:**

Accept (poster)

**Comment:**

The paper analyzes prior ways of conditioning multi-view transformers with camera information and proposes an extension to prior conditionings that incorporates camera intrinsics. The proposed conditionings (at attention and token level) show consistent improvements across methods.

The reviewers appreciated the analysis and proposed conditioning as a sound & elegant extension to prior work. They noted the consistent and, for some conditions, significant improvements. There were initial doubts whether the attention-level conditioning would provide a benefit over token-level conditioning and what the reason for the strong performance of the latter would be. Further concerns were raised about missing experimental conditions and the paper being a mere extension over prior work. But the extensive rebuttal provided additional results and effectively clarified all remaining critical issues. The reviewers and the AC unanimously agree that the presented approach is simple, novel, well motivated, and clearly shown to provide consistent improvements over prior work.